# GGFlow: A Graph Flow Matching Method with Efficient Optimal Transport

**Xiaoyang Hou**[*]    *houxiaoyang22s@ict.ac.cn*
*Institute of Computing Technology*
*University of the Chinese Academy of Sciences*

**Tian Zhu**[*]    *zhutian22s@ict.ac.cn*
*Institute of Computing Technology*
*University of the Chinese Academy of Sciences*

**Milong Ren**    *renmilong21b@ict.ac.cn*
*Institute of Computing Technology*
*University of the Chinese Academy of Sciences*

**Dongbo Bu**    *dbu@ict.ac.cn*
*Institute of Computing Technology*
*University of the Chinese Academy of Sciences*
*Central China Research Institute for Artificial Intelligence Technologies*

**Chunming Zhang**    *zhangchunming@ncic.ac.cn*
*Institute of Computing Technology*
*Western Institute of Computing Technology*
*Phil Rivers Technology*

**Xin Gao** [†]    *xin.gao@kaust.edu.sa*
*King Abdullah University of Science and Technology*

**Shiwei Sun** [†]    *dwsun@ict.ac.cn*
*Institute of Computing Technology*
*University of the Chinese Academy of Sciences*
*Western Institute of Computing Technology*

**Reviewed on OpenReview:** *https://openreview.net/forum?id=K8RlXtMgzo*

## Abstract

Generating graph-structured data is crucial in various domains but remains challenging due to the complex interdependencies between nodes and edges. To enhance generation performance, we propose GGFlow, a discrete flow matching generative model with an efficient optimal transport for graph structures and it incorporates an edge-augmented graph transformer. Additionally, GGFlow introduces a novel goal-guided generation framework to control the generative trajectory of our model towards desired properties. GGFlow demonstrates superior performance on both unconditional and conditional generation tasks, outperforming existing baselines and underscoring its effectiveness and potential for wider application. Code: `https://github.com/Xiaoyang878/GGFlow`

## 1 Introduction

Graph structural data generation has become critically important across various domains, including social networks (Grover et al., 2019), drug design (Bilodeau et al., 2022), and neural architecture search (NAS) (Lee et al., 2020). Effective modeling of the intrinsic joint distribution and accurate description of topological structures of graphs are essential for these applications. Deep generative models have increasingly demonstrated success in graph generation by effectively modeling the complex structural properties of graphs. These models are typically categorized into autoregressive and one-shot types. Autoregressive models, such as GraphRNN

---

[*]Contributed equally: Author order is randomized and can be adjusted as needed for individual purposes.

[†]Correspondence should be addressed to Shiwei Sun (dwsun@ict.ac.cn) and Xin Gao (xin.gao@kaust.edu.sa)

(You et al., 2018), generate graphs sequentially, often overlooking the interdependencies among all graph components. In contrast, one-shot methods generate entire graphs in a single step, more effectively capturing the joint distribution (Ma et al., 2018; Luo et al., 2023; Niu et al., 2020).

Diffusion models have shown great promise and achieved significant performance in various domains (Ho et al., 2020; Song et al., 2020; Ho et al., 2022). In the context of graph generation, diffusion models have been adopted to enhance generative capacity. EDP-GNN and GDSS are among the first to utilize diffusion models for graph generation, adding continuous Gaussian noise to adjacency matrices and node types, which may lead to invalid graph structures (Niu et al., 2020; Jo et al., 2022b). Due to the inherent sparsity and discreteness of graph structures, GSDM enhances model fidelity by introducing Gaussian noise within a continuous spectrum space of the graph, DiGress and PPGN apply discrete diffusion models for graphs (Luo et al., 2023; Vignac et al., 2022; Austin et al., 2021; Haefeli et al., 2022; Huang et al., 2023).

Flow matching generative models also offer a compelling alternative. As these methods transform the training paradigm from SDEs to a more stable interpolation method, they enable more efficient sampling via ordinary differential equations (ODEs) (Lipman et al., 2022; Song et al., 2024; Yim et al., 2023). Furthermore, the application of optimal transport (OT) has been explored to straighten marginal probability paths, leading to reduced training variance and accelerated sampling (Bose et al., 2023; Tong et al., 2023; Klein et al., 2024; Pooladian et al., 2023). Nevertheless, the integration of OT into graph-based systems often encounters substantial computational hurdles due to the inherent complexity of computing OT metrics for discrete structures (Chen et al., 2020b; Petric Maretic et al., 2019).

In this paper, we introduce GGFlow, a novel generative model that leverages discrete flow matching techniques with an efficient optimal transport coupling in graph generation. The model preserves graph sparsity and permutation invariance, which is essential for realistic graph generation. Additionally, GGFlow employs a goal-guided framework using reinforcement learning for conditional generation. GGFlow achieves state-of-the-art results in both unconditional and conditional graph and molecule generation tasks and surpasses existing methods. Its effectiveness in conditional generation tasks underscores the practical impact of our approach.

Our contribution can be summarized as:

- GGFlow introduces the first discrete flow matching generative model with an efficient optimal transport for graph data, and it also incorporates an edge-augmented graph transformer to enhance generation tasks further.

- GGFlow proposes a novel guidance framework using reinforcement learning to control probability flow during graph generation, targeting specific properties.

- GGFlow demonstrates state-of-the-art performance in various unconditional and conditional graph generation tasks, consistently outperforming existing methods across diverse graph types and complexities.

## 2 Related Work

### 2.1 Flow Matching and Diffusion Models

Diffusion models have gained widespread popularity in various fields, including computer vision, natural language processing, and biological sciences, demonstrating notable success in generative tasks (Ho et al., 2020; Song et al., 2020; Watson et al., 2023; Ingraham et al., 2023; Liu et al., 2024a; Ren et al., 2024; Zhu et al., 2024). Recently, flow matching generative models have emerged as a more efficient and stable alternative (details in Appendix A.1), improving generative performance (Lipman et al., 2022; Song et al., 2024; Campbell et al., 2024). Some approaches further enhance performance by incorporating optimal transport. The generative processes of these models are summarized in Figure 1.

Previous works (Campbell et al., 2024; Gat et al., 2024) extended flow matching to discrete spaces, while Eijkelboom et al. (2024) applied variational flow matching to graphs, but without adequately addressing key graph-specific properties such as adjacency matrix sparsity. GGFlow tackles these challenges by introducing

a discrete flow matching model with an efficient optimal transport tailored for graph data. Furthermore, we propose a novel framework for guiding the generative process, enhancing its practical applicability.

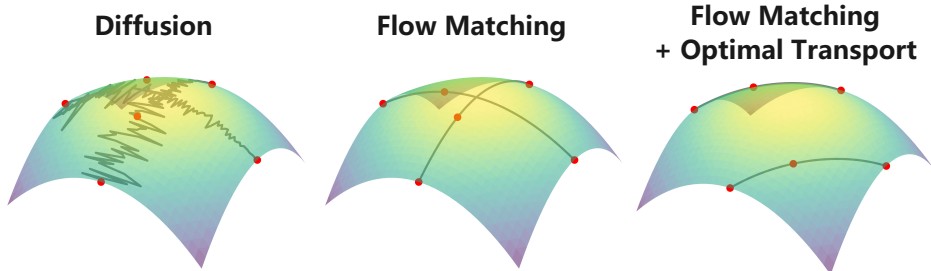

Figure 1: Illustration of generative trajectories using different methods. The generative trajectories are learned by the diffusion model (left), flow matching model (center), and flow matching model with optimal transport (right).

## 2.2 Graph Generative Models

Graph generative models are typically categorized into two main types: autoregressive and one-shot models. Autoregressive models generate graphs sequentially (Wang et al., 2018; You et al., 2018; Jin et al., 2018; Shi et al., 2019; Luo et al., 2021; Kong et al., 2023). While effective, these models are often computationally expensive and fail to account for permutation invariance, a crucial property for graph data, resulting in potential inefficiencies. In contrast, one-shot models aim to capture the distribution of all graph components simultaneously (De Cao & Kipf, 2018; Ma et al., 2018; Zang & Wang, 2020), better reflecting the inherent interactions within graphs. Despite the advantages, diffusion-based one-shot models (Niu et al., 2020; Jo et al., 2022b; Vignac et al., 2022; Chen et al., 2023; Bergmeister et al., 2023; Luo et al., 2023; Haefeli et al., 2022; Yan et al., 2023; Jang et al., 2023; Madeira et al., 2024; Bergmeister et al., 2024; Chen et al., 2023; Minello et al., 2024; Zhao et al., 2024; Xu et al., 2024) show promising results in downstream tasks. GGFlow employs a discrete flow-matching generative model, achieving superior generative performance. More comparisons with recent works are presented in Appendix B.

## 3 Background

Discrete Flow Matching (DFM) generative models extend the flow matching paradigm to generate data samples within a discrete domain. We consider a discrete variable $x_t$ taking values in a finite state space denoted by $\mathcal{N} = \{1, \dots, N\}$. The marginal distribution of $x_t$ is represented by the probability vector $p_t$, satisfying $\sum_{i=1}^{N} p_t(x_t = i) = 1$. The process interpolates between a predefined noise prior $p_0 = p_{\text{ref}}$ (typically a uniform distribution or mask distribution) and the target data distribution $p_1 = p_{\text{data}}$ (Campbell et al., 2022; Gat et al., 2024). The marginal probability path is defined as:

$$p_t(x) = \sum_{x_0, x_1 \in \mathcal{N}} p_t(x|x_1, x_0)\pi(x_1, x_0), \tag{1}$$

where the training pairs are sampled from the joint distribution $\pi(x_1, x_0)$. In the simplest scenario, $\pi$ is modeled as an independent coupling, i.e., $\pi(x_1, x_0) = p_{\text{data}}(x_1)p_{\text{ref}}(x_0)$.

Following Campbell et al. (2024), the conditional probability path $p_{t|1}(x_t|x_1, x_0)$ is defined via linear interpolation. Given a data sample $x_1$ and a prior sample $x_0$, the path and independent coupling are given by:

$$p_{t|1}(x_t|x_1, x_0) = t\delta(x_t, x_1) + (1-t)\delta(x_t, x_0), \tag{2}$$

$$\pi(x_1, x_0) = p_{\text{data}}(x_1)p_0(x_0), \tag{3}$$

where $\delta$ denotes the Kronecker delta function.

For the sampling procedure, DFM leverages a Continuous-Time Markov Chain (CTMC) formulation. The CTMC is characterized by an initial distribution $p_0$ and a time-dependent probability rate matrix (analogous to a velocity field) $u_t(\cdot, \cdot) \in \mathbb{R}^{N \times N}$. The transition dynamics over a small step $\Delta t$ are defined as:

$$p_{t+\Delta t|t}(x_{t+\Delta t}|x_t) = \delta(x_{t+\Delta t}, x_t) + u_t(x_t, x_{t+\Delta t})\Delta t, \tag{4}$$

where $u_t(x_t, x_{t+\Delta t})$ represents the transition rate between states. Intuitively, $u_t(x_t, x_{t+\Delta t})\Delta t$ yields the probability that a transition from state $x_t$ to state $x_{t+\Delta t}$ will occur in the next infinitesimal time step $\Delta t$. To ensure proper normalization such that $\sum_{x_{t+\Delta t}} p_{t+\Delta t|t}(x_{t+\Delta t}|x_t) = 1$, the diagonal entries are defined as $u_t(x_t, x_t) = -\sum_{k \neq x_t} u_t(x_t, k)$, while off-diagonal entries are non-negative. Consequently, the marginal distribution evolves according to the Kolmogorov forward equation:

$$\partial_t p_t = u_t^T p_t \tag{5}$$

Specifically, DFM relies on a conditional rate matrix $u_t(\cdot, \cdot|x_1) \in \mathbb{R}^{N \times N}$ to govern the denoising process. Under mild assumptions, Campbell et al. (2024) derive a closed-form solution for this valid conditional rate matrix. For off-diagonal entries ($x_t \neq x_{t+\Delta t}$), this is defined as:

$$u_t^*(x_t, x_{t+\Delta t}|x_1) = \frac{\text{ReLU}[\partial_t p_{t|1}(x_{t+\Delta t}|x_1) - \partial_t p_{t|1}(x_t|x_1)]}{\mathbf{Z}_t p_{t|1}(x_t|x_1)}, \tag{6}$$

where $\mathbf{Z}_t = |\{x : p_{t|1}(x|x_1) > 0\}|$ is a normalization constant. The marginal velocity field is then obtained via expectations, $u_t(x_t, x_{t+\Delta t}) = \mathbb{E}_{p_{1|t}(x_1|x_t)}[u_t^*(x_{t+\Delta t}, x_t|x_1)]$, which can be substituted into Equation (4) for generation.

The discrete flow matching formulation introduced by Campbell et al. (2024) was recently adapted for discrete graph generation in DeFoG (Qin et al., 2024). DeFoG leverages discrete flow matching with independent coupling to model the generation of each node and edge independently, successfully demonstrating the benefits of training-sampling disentanglement in the graph domain. However, the utility of incorporating Optimal Transport (OT) coupling into this discrete graph flow matching framework—specifically concerning its impact on sample quality, and the smoothness of the learned flow—remains a critical, underexplored avenue. To address this critical gap and fully harness the theoretical advantages of OT, we introduce our method, GGFlow.

## 4 Methods

In this section, we present our methodology, GGFlow. Section 4.1 outlines the discrete flow matching method for graph generation. Section 4.2 covers an efficient optimal transport for graph flow matching. Section 4.3 introduces GraphEvo, our neural network for graph generation. Section 4.4 examines the permutation properties of GGFlow, and Section 4.5 discusses goal-guided graph generation using reinforcement learning.

### 4.1 Discrete Flow Matching for Graph Generation

A graph $G = (V, E)$, where $V$ and $E$ denote the sets of nodes and edges, has a distribution denoted by $p(G) \triangleq (p^V(V), p^E(E))$. The attribute spaces for nodes and edges are $\mathcal{V}$ and $\mathcal{E}$, with cardinalities $n$ and $m$, respectively. The attributes of node $i$ and edge $ij$ are denoted by $v_i \in \mathcal{V}$ and $e_{ij} \in \mathcal{E}$, so the node and edge probability mass functions (PMF) are $p(v_i = a)$ and $p(e_{ij} = b)$ where $a \in \{1, \ldots, n\}$ and $b \in \{1, \ldots, m\}$.

We formulate the noising and generation trajectory by treating the attributes of each node and edge as independent random variables. Let $x$ be a generic variable $x \in \mathcal{V} \cup \mathcal{E}$. Under the assumption of independence across all components, the distribution of the noisy graph $G^t$ at time $t$ is defined as the product of the individual component probabilities:

$$p_t(G^t) \triangleq (p_t^V(V^t), p_t^E(E^t)) \triangleq \prod_i p(v_i^t) \prod_{j,k} p(e_{jk}^t) = \prod_{x \in \mathcal{V} \cup \mathcal{E}} p(x^t). \tag{7}$$

where $G$ denotes the set of all node and edge variables, and $p_t(x^t)$ is the PMF of a single variable $x$ at time $t$.

The node and edge embeddings in the graph are given by matrices $\mathbf{V} \in \mathbb{R}^{N \times d_v}$ and $\mathbf{E} \in \mathbb{R}^{N \times N \times d_e}$, respectively, where $N$ is the number of nodes, $d_v$ is the hidden dim of node embeddings and $d_e$ is the hidden dim of edge embeddings. We denote the transpose of matrix $\mathbf{A}$ as $\mathbf{A}^*$ and $\mathbf{A}^t$ represents the state of matrix $\mathbf{A}$ at time $t$. We use discrete flow matching to model the graph generation process.

**Source and target distribution** GGFlow aims to transform prior distribution $G^0 \sim p_{\text{ref}}$ to target data distribution $G^1 \sim p_{\text{data}}$. The training data $(G^0, G^1)$ are sampled from a joint distribution $\pi(G^0, G^1)$, satisfying the marginals constraints $p_{\text{ref}} = \sum_{G^1} \pi(G^0, G^1), p_{\text{data}} = \sum_{G^0} \pi(G^0, G^1)$.

To account for graph sparsity, the prior distribution $p_{\text{ref}} = (p_{\text{ref}}^V, p_{\text{ref}}^E)$ is designed to approximate the true data distribution closely. To ensure the permutation invariance of the model, the priors are structured as products of single distributions for all nodes and edges: $\prod_i v_i \times \prod_{ij} e_{ij}$ (Vignac et al., 2022). Further details on the prior can be found in Appendix C.1.

**Probability path and coupling** We define a probability path $p_t(G^t)$ that interpolates between source distribution $p_{\text{ref}}$ and target distribution $p_{\text{data}}$ i.e. $p_0 = p_{\text{ref}}$ and $p_1 = p_{\text{data}}$. The marginal probability path is given by:

$$p_t(G^t) = \sum_{(G^0, G^1) \sim \pi} p_{t|1}(G^t | G^0, G^1) \pi(G^0, G^1), \tag{8}$$

where

$$
\begin{aligned}
p_{t|1}(G|G^0, G^1) &\triangleq t\delta\{G, G^1\} + (1-t)\delta\{G, G^0\} \\
&\triangleq \left( t\delta\{V, V^1\} + (1-t)\delta\{V, V^0\}, t\delta\{E, E^1\} + (1-t)\delta\{E, E^0\} \right) \\
&\triangleq \prod_{x \in \mathcal{V} \cup \mathcal{E}} (t\delta\{x, x^1\} + (1-t)\delta\{x, x^0\})
\end{aligned}
$$

$\delta$ is the Kronecker delta, indicating equality of the indices, the sampling procedure is independent for each node and edge. Given the sparsity of both the prior and data distributions, we can infer that the intermediate distribution is similarly sparse, aiding model training.

Critically, we employ the Optimal Transport (OT) map as the coupling between the marginals. Compared to the independent coupling used in previous works (Campbell et al., 2024; Qin et al., 2024), OT provides a generalized and more structured coupling that enhances the quality of conditional samples (Tong et al., 2023; Gat et al., 2024) (detailed in Section 4.2).

We define a probability velocity field:

$$u_t(G^t, G) \triangleq (u_t^V(V^t, V), u_t^E(E^t, E)) \triangleq \prod_{x \in \mathcal{V} \cup \mathcal{E}} u_t(x^t, x)$$

for GGFlow, which generates the probability path from Equation 8. The probability velocity field $u_t(G^t, G)$ is derived from the conditional probability velocity field $u_t(G^t, G|G^0, G^1)$, and can be expressed as:

$$u_t(G^t, G) = \sum_{(G^0, G^1)} u_t(G^t, G|G^0, G^1) p_t(G^0, G^1|G), \tag{9}$$

$$p_t(G^0, G^1|G^t) = p_{1|t}(G^1|G^t, G^0) \frac{p_t(G^t|G^0, G^1)\pi(G^0, G^1)}{\sum_{G^0, G^1} p_t(G^t|G^0, G^1)\pi(G^0, G^1)}. \tag{10}$$

GGFlow chooses the conditional marginal probability $u_t(G^t, G|G^0, G^1)$ as:

$$u_t(G^t, G|G^0, G^1) \triangleq \prod_{x \in \mathcal{V} \cup \mathcal{E}} u_t(x^t, x|x^0, x^1) = \prod_{x \in \mathcal{V} \cup \mathcal{E}} \left( \frac{\delta\{x^t, x^0\}}{\mathbf{Z}_t(1-t)} \left( 1 + \delta\{x, x^1\} - \delta\{x, x^0\} \right) \right), \tag{11}$$

where $\mathbf{Z}_t = |\{x : p_t(x|x^0, x^1) > 0, x \in \mathcal{V} \cup \mathcal{E}\}|$. More details about the choice of the conditional vector field are provided in Appendix C.2.

**Training objective** Given the intractability of the posterior distribution $p_{1|t}(G^1|G^t, G^0)$, we approximate it as $\hat{p}_{1|t}(G^1|G^t, G^0)$ using neural network, GraphEvo, as detailed in Section 4.3. The training objective is formulated as:

$$\mathcal{L} = \mathbb{E}_{p_{\text{data}}(G^1)\mathcal{U}(t;0,1)\pi(G^0,G^1)p_{t|1}(G^t|G^0,G^1)}[\log \hat{p}_{1|t}(G^1|G^t, G^0)], \tag{12}$$

where $\mathcal{U}(t; 0, 1)$ is a uniform distribution on $[0, 1]$.

**Sampling Procedure** In the absence of the data distribution $G^1$ during sampling, we can simplify the generative process $p_{t+\Delta t|t}(G^{t+\Delta t}|G^t, G^0)$ without the calculation of the full expectation over the conditional vector field $u_t(G^t, G^{t+\Delta t}|G^0, G^1)$:

$$p_{t+\Delta t|t}(G^{t+\Delta t}|G^t, G^0) = \mathbb{E}_{\hat{p}_{1|t}(G^1|G^t,G^0)}[\delta(G^{t+\Delta t}, G^t) + u_t(G^t, G^{t+\Delta t}|G^0, G^1)\Delta t]$$
$$= \sum_{G^1} p_{t+\Delta t|t}(G^{t+\Delta t}|G^t, G^1, G^0)\hat{p}_{1|t}(G^1|G^t, G^0). \tag{13}$$

This expectation can be sampled efficiently using a two-step procedure (Algorithm 1): (1) first, we sample the endpoint $\hat{G}^1 \sim \hat{p}_{1|t}(G^1|G^t, G^0)$ using the approximate posterior, and (2) then, we sample the next state $G^{t+\Delta t}$ conditioned on $\hat{G}^1$. The conditional sampling transition is given by:

$$G^{t+\Delta t} \sim p_{t+\Delta t|t}(G^{t+\Delta t}|G^t, G^1, G^0) = \delta\{G^{t+\Delta t}, G^t\} + u_t(G^t, G^{t+\Delta t}|G^0, \hat{G}^1)\Delta t,$$

where the term $u_t(G^t, G^{t+\Delta t}|G^0, \hat{G}^1)\Delta t$ represents the conditional probability of a transition from $G^t$ to $G^{t+\Delta t}$ in the infinitesimal time step $\Delta t$, given the initial condition $G^0$ and the sampled endpoint $\hat{G}^1$. Further details on the sampling and training procedures are provided in Algorithms 1 and 3.

---

**Algorithm 1** Sampling Procedure of GGFlow

---

**Require:** $t = 0, G^0 \sim (p_V^{\text{ref}}, p_E^{\text{ref}}), u_t(G^t, G|G^0, G^1), N_{\text{steps}}$
1: $\Delta t = 1/N_{\text{steps}}$
2: **for** $n \in \{0, \ldots, N_{\text{steps}} - 1\}$ **do**
3:     $\hat{p}_{1|t}(G^1|G^t, G^0) = \text{GraphEvo}(G^t, t)$
4:     $\hat{G}^1 \sim \hat{p}_{1|t}(\hat{G}^1|G^t, G^0)$
5:     // Sampling from the conditional velocity field
6:     $G^{t+\Delta t} \sim p_{t+\Delta t|t}(G^{t+\Delta t}|G^t, G^1, G^0) = \delta\{G^{t+\Delta t}, G^t\} + u_t(G^t, G^{t+\Delta t}|G^0, \hat{G}^1)\Delta t$
7:     $t = t + \Delta t$
8: **end for**
9: **return** $G^1 = (V^1, E^1)$

---

## 4.2 Efficient Optimal transport for graph flow matching

Optimal transport (OT) has been effectively applied to flow matching generative models in continuous variable spaces, to improve generative performance (Tong et al., 2023; Bose et al., 2023; Song et al., 2024). To generalize this for graphs, we set the joint distribution $\pi(G^0, G^1)$ as a coupling to the 2-Wasserstein OT map, which minimizes the 2-Wasserstein distance between $p_{\text{ref}}$ and $p_{\text{data}}$. Let $\mathcal{G}$ denote the space of graphs. The optimal transport plan $\pi^*(G^0, G^1)$ is defined as the minimizer of the expected cost (Bookstein et al., 2002):

$$\pi(G^0, G^1) = \arg\inf_{\pi \in \Phi(p_0, p_1)} \int_{\mathcal{G} \times \mathcal{G}} H(G^0, G^1)\mathrm{d}\pi(G^0, G^1), \tag{14}$$

where

$$H(G^0, G^1) = \sum_i \delta(v_i^0, v_i^1) + \lambda \sum_{i,j} \delta(e_{ij}^0, e_{ij}^1). \tag{15}$$

where $\Phi(p_0, p_1)$ represents the set of all joint probability measures (couplings) on $\mathcal{G} \times \mathcal{G}$ whose marginals are $p_0$ and $p_1$, respectively, where $G^K = (V^K = \{v_i^K\}, E^K = \{e_{ij}^K\}_{ij})$, $K = 0, 1$.

The practical application of OT to large datasets is computationally intensive, often requiring cubic time complexity and quadratic memory (Tong et al., 2020; Villani, 2009). To address these challenges, we use a minibatch approximation of OT (Fatras et al., 2021). A detailed analysis of optimal transport during the training procedure is provided in Appendix E.

### 4.3 GraphEvo: Edge-augmented Graph Transformer

Our neural network, GraphEvo, is tasked with predicting the necessary factorized posterior distribution, $\hat{p}_{1|t}(G^1|G^t, G^0)$. Given the intermediate graph state $G^t = \{V^t, E^t\}$ and the time step $t$, GraphEvo outputs the predicted distribution for the final state $G^1$:

$$\hat{p}_{1|t}(G^1|G^t) = (\hat{p}_{1|t}^V(V^1|V^t), \hat{p}_{1|t}^E(E^1|E^t)).$$

For theoretical consistency with the flow matching framework, the full posterior is denoted as $\hat{p}_{1|t}(G^1|G^t, G^0)$, acknowledging the implicit dependency on the initial noise state $G^0$ via the time-dependent path. The model itself, however, primarily operates on $G^t$ and $t$.

Since structural information and edge relations are as critical as node attributes for capturing complex graph topologies and are known to enhance link generation tasks (Hussain et al., 2024; Hou et al., 2024; Jumper et al., 2021), GraphEvo is built upon a specialized Transformer architecture. To effectively capture non-local dependencies and higher-order structural motifs, we extend the transformer with a triangle attention mechanism for dynamic edge updates. Furthermore, we augment the input features of the graph, represented by node and edge embedding matrices $\mathbf{V} \in \mathbb{R}^{N \times d_v}$ and $\mathbf{E} \in \mathbb{R}^{N \times N \times d_e}$ (where $N$ is the number of nodes, $d_v$ is the hidden dim of node embeddings and $d_e$ is the hidden dim of edge embeddings.), with rich global auxiliary graph features $\mathbf{y} \in \mathbb{R}^{1 \times h}$, such as cycles and the number of connected components (Vignac et al., 2022). Here, $h$ is the dimension of the auxiliary features. This comprehensive approach enables GraphEvo to efficiently and accurately model the joint distribution of all graph components, as detailed in Algorithm 2.

### 4.4 Permutation Property Analysis

Graphs are invariant to random node permutations, and GGFlow preserves this property. To ensure permutation invariance, we analyze the permutation properties of our neural network, training objectives, and conditional probabilities path. First, we analyze the permutation invariance of the training objectives (Vignac et al., 2022). Since the source and target distributions are permutation invariant, the independent coupling also exhibits this invariance. Our optimal transport map, derived from Equation 14, similarly demonstrates invariance to identical permutations. Further clarifications regarding optimal transport can be found in Appendix C.4.

**Theorem 1.** *If the distributions $p(G^0)$ and $p(G^1)$ are permutation invariant, and the cost function maintains invariance under identical permutations, i.e., $H(G^0, G^1) = H(\sigma G^0, \sigma G^1)$ for any permutation $\sigma$, then the optimal transport map $\pi$ also exhibits invariance under identical permutations, such that $\pi(G^0, G^1) = \pi(\sigma G^0, \sigma G^1)$.*

Proof of this theorem can be found in Appendix C.4. To ensure that the generated graph retains its identity under random permutations, the generated distribution must remain exchangeable, and GraphEvo must be permutation equivariant.

**Proposition 1.** *The distribution generated by the conditional flow is exchangeable with respect to nodes and graphs, i.e. $p(G) = (p^V(V), p^E(E)) = (p^V(\sigma^*V), p^E(\sigma^*E\sigma))$, where $\sigma$ is a permutation operator.*

**Proposition 2.** *GraphEvo is permutation equivariant.*

The proofs of Proposition 1 and 2 are provided in Appendix C.3 and Appendix D.1, respectively.

---

**Algorithm 2** Architecture of GraphEvo

---

**Require:** $G^t, t, N_{\text{layer}}$
1: **// Initialization Phase**
2: $V^t, E^t \leftarrow G^t$
3: $\mathbf{y} \leftarrow \text{ExtractFeature}(G^t), \mathbf{t} \leftarrow \text{TimeEmbedding}(t)$
4: $\mathbf{y} \leftarrow \mathbf{y} + \mathbf{t}$
5: $\mathbf{V}, \mathbf{E}, \mathbf{y} \leftarrow \text{Linear}(V^t), \text{Linear}(E^t), \text{Linear}(\mathbf{y})$
6: **// Triangle Attention Phase**
7: **for** $l = 0, 1, \ldots, N_{\text{layer}}$ **do**
8:    // Definition: Triangle-attention Mechanism Computation
9:    *// 1. Node Attention*
10:    $\mathbf{Q_v}, \mathbf{K_v}, \mathbf{V_v} \leftarrow \text{Linear}(\mathbf{V})$
11:    $\mathbf{Y_v} \leftarrow \frac{\mathbf{Q_v} \times \mathbf{K_v}}{\sqrt{dx}}$
12:    *// 2. Incorporate edge features to self-attention scores*
13:    $\mathbf{Y_v} \leftarrow \text{FiLM}(\mathbf{Y_v}, \mathbf{E})$
14:    $\mathbf{E_{tmp}} \leftarrow \mathbf{Y_v}$
15:    *// 3. Edge Triangle Attention*
16:    $\mathbf{Q_e}, \mathbf{K_e}, \mathbf{V_e}, \mathbf{b}, \mathbf{g} \leftarrow \text{Linear}(\mathbf{E_{tmp}})$
17:    $\mathbf{Y_e} \leftarrow \frac{\mathbf{Q_e} \times \mathbf{K_e}}{\sqrt{dx}} + \mathbf{b}$
18:    $\mathbf{E}' \leftarrow \mathbf{Y_e} * \mathbf{V_e} * \text{sigmoid}(\mathbf{g})$
19:    $\mathbf{E}' \leftarrow \text{Linear}\Big(\text{FiLM}(\mathbf{E}', \mathbf{y})\Big)$    // Incorporate global to edge
20:    *// 4. Update Node and Global features*
21:    $\mathbf{V}' \leftarrow \mathbf{Y_v} * \mathbf{V_v}$
22:    $\mathbf{V}' \leftarrow \text{Linear}\Big(\text{FiLM}(\mathbf{V}', \mathbf{y})\Big)$    // Incorporate global to node
23:    $\mathbf{y}' \leftarrow \text{Linear}\Big(\text{Linear}(\mathbf{y}) + \text{PNA}(\mathbf{V}) + \text{PNA}(\mathbf{E})\Big)$
24:    // Feed-Forward & Residual Connections
25:    $\mathbf{V} \leftarrow \text{ReLU}\Big(\text{LayerNorm}(\mathbf{V} + \text{Dropout}(\mathbf{V}'))\Big)$
26:    $\mathbf{E} \leftarrow \text{ReLU}\Big(\text{LayerNorm}(\mathbf{E} + \text{Dropout}(\mathbf{E}'))\Big)$
27:    $\mathbf{y} \leftarrow \text{ReLU}\Big(\text{LayerNorm}(\mathbf{y} + \text{Dropout}(\mathbf{y}'))\Big)$
28: **end for**
29: **// Prediction Phase**
30: $\hat{p}_{1|t}^V(V^1|V^t, V^0), \hat{p}_{1|t}^E(E^1|E^t, E^0), \mathbf{y} \leftarrow \text{Linear}(\mathbf{V}), \text{Linear}(\mathbf{E}), \text{Linear}(\mathbf{y})$
31: $\hat{p}_{1|t}(G^1|G^t, G^0) = (\hat{p}_{1|t}^V(V^1|V^t, V^0), \hat{p}_{1|t}^E(E^1|E^t, E^0))$
32: **return** $\hat{p}_{1|t}(G^1|G^t, G^0)$

---

### 4.5 Goal-Guided Framework for Conditional Generation

For practical applications such as drug discovery, we propose a goal-guided framework for discrete flow matching, employing reinforcement learning (RL) to guide graph flow matching models for non-differentiable objectives. The goal of the guidance method is to map the noise distribution $p_0$ to a preference data distribution $p_1^*$ using a reward function $\mathcal{R}(G^t, t)$.

**Formulation**  We cast the diffusion process as a Markov Decision Process (MDP). Concretely, our formulation builds upon prior work (Wallace et al., 2024). We adopt and extend this trajectory-level RL formulation to discrete flow-matching. The state at decision epoch $t$ is the flow-matching representation $\boldsymbol{s}_t = (G^t, t)$, and the action space $\boldsymbol{a}_t$ comprises all realizable successor states $G^{t+\Delta t}$ at the next time $t + \Delta t$. $p_{t+\Delta t|t}(G^{t+\Delta t}|G^t)$ is the transition dynamics and policy network $\mu(\mathbf{a}_t|\mathbf{s}_t)$.

**Reward function**  To align with practical scenarios where only the final outcome can be evaluated, we employ a sparse reward structure. The per-step reward function $\mathcal{R}(G^t, t)$ is zero for all intermediate steps, and the final reward is defined solely on the generated sample $G^1$:

$$\mathcal{R}(G^t, t) = r(G^1)\mathbb{I}[t = 1]. \tag{16}$$

where $r(G^1)$ is the objective function (e.g., predicted molecular property). The RL agent seeks to maximize the total cumulative reward $\mathcal{R} = \sum_{t \in [0,1]} \mathcal{R}(G^t, t)$, which simplifies to $\mathcal{R} = r(G^1)$.

**Exploration**  To enable exploration, we additionally introduce a temperature parameter $\tau$ for the policy network during sampling, allowing the model to explore a broader space at higher temperatures:

$$\mu(\mathbf{a}_t|\mathbf{s}_t) = p_{t+\Delta t|t}^{\tau}(G^{t+\Delta t}|G^t) = \frac{1}{\tau}(\delta\{\cdot, G^t\} + u_t(\cdot, G^t)\Delta t). \tag{17}$$

As detailed in Section 4.1, this transition probability $p_{t+\Delta t|t}^{\tau}(G^{t+\Delta t}|G^t)$ is efficiently sampled using a two-step procedure: first sampling the $\hat{G}^1$ endpoint, and then sampling the next state $G^{t+\Delta t}$ conditioned on the endpoint. Meanwhile, we demonstrate in the 2-D Gaussian case that this approach is able to explore a significantly larger region of the space (Appendix F).

**Training**  The goal of RL training is to maximize the reward function. To prevent overfitting to the reward preference distribution, we add a Kullback–Leibler (KL) divergence term between the Reinforcement learning fine-tuned model $p_\theta^{RL}(\cdot)$ and pre-trained model $p_\theta(\cdot)$ (Ouyang et al., 2022).

We adapt the policy gradient method established in Wallace et al. (2024) to fine-tune our network parameters $\theta$ for goal-guided generation. The policy $\mu(\mathbf{a}_t|\mathbf{s}_t)$ is implicitly defined by the model's prediction of the final state $p_\theta^{\mathrm{RL}}(G^1|G^t, G^0)$.

Similar to previous work (Liu et al., 2024b), under our definitions of reinforcement learning, the training objective can be written as:

$$\mathcal{L}_{RL} = -\mathbb{E}_{p_\theta(G^{0:t:1})}[\alpha\mathcal{R}(G^t, t)\sum_{t=0}^{1}\log p_\theta^{RL}(G^1|G^t, G^0) - \beta\sum_{t=0}^{1}\mathrm{KL}(p_\theta^{RL}(G^1|G^t, G^0)||p_\theta(G^1|G^t, G^0))] \tag{18}$$

where $p_\theta(G^{0:t:1})$ represents $p_{\mathrm{data}}(G^1)\mathcal{U}(t; 0, 1)\pi(G^0, G^1)p_t(G^t|G^0, G^1)$.

Using this optimization objective, we fine-tune the pre-trained flow matching model to generate data following the preference distribution. By integrating optimal transport, we optimize the pairing of prior data and high-reward training data (Chen et al., 2020a). The pseudo-code for the guided GGFlow training is provided in Algorithm 4 and a toy example is shown in Appendix F.

## 5 Experiments

To validate the performance of our method, we compare GGFlow with state-of-the-art graph generative baselines on generic graph generation and molecule generation, over several benchmarks in Section 5.1 and Section 5.2, respectively. The ability of GGFlow to perform conditional generation is analyzed in Section 5.3. Finally, we conduct detailed ablation studies presented in Section 5.4.

## 5.1 Generic Graph Generation

We evaluated GGFlow on five generic graph generation benchmarks of varying sizes: Ego-small, Community-small, Grid, Planar and Enzymes. We employ the same train/test split as GraphRNN (You et al., 2018), utilizing 80% of each dataset for training and the remaining for testing. We compared GGFlow's performance against well-known autoregressive models: DeepGMG (Li et al., 2018), GraphRNN (You et al., 2018), GraphAF (Shi et al., 2019), and GraphDF (Luo et al., 2021) and one-shot models: GraphVAE (Simonovsky & Komodakis, 2018), GNF (Liu et al., 2019), EDP-GNN (Niu et al., 2020), GDSS (Jo et al., 2022a), DiGress (Vignac et al., 2022), GRASP (Minello et al., 2024), GSDM (Luo et al., 2023), GruM (Jo et al., 2024), and SwinGNN (Yan et al., 2023). Consistent with previous studies, we generated an equal number of graphs as the test set to compare distributions of graph statistics, including degree distribution (Deg.), clustering coefficient (Clus.), and the frequency of 4 node orbits (Orbit). Detailed descriptions of datasets, baselines, and metrics are provided in Appendix H.

Table 1 presents our results, showing that GGFlow achieves superior performance across most metrics. Additionally, GGFlow demonstrates comparable performance compared to state-of-the-art models in generating large graphs on the Grid dataset. These findings underscore the effectiveness of GGFlow at capturing the local characteristics and data distributions of graphs. Additional metrics and dataset experimental results are included in Appendix G, and we visualize the generated graphs in Appendix J.

Table 1: Generation results on the generic graph datasets. Results are the means of 3 different runs. The best results and the second-best results are marked **bold** and bold.

| Method | Ego-small | | | | Community-small | | | | Grid | | | | Step |
|---|---|---|---|---|---|---|---|---|---|---|---|---|---|
| | Deg. | Clus. | Orbit | Avg. | Deg. | Clus. | Orbit | Avg. | Deg. | Clus. | Orbit | Avg. | |
| Training Set | 0.014 | 0.022 | 0.004 | 0.013 | 0.003 | 0.009 | 0.001 | 0.005 | 0.000 | 0.000 | 0.000 | 0.000 | - |
| DeepGMG | 0.040 | 0.100 | 0.020 | 0.053 | 0.220 | 0.950 | 0.400 | 0.523 | - | - | - | - | - |
| GraphRNN | 0.090 | 0.220 | 0.003 | 0.104 | 0.080 | 0.120 | 0.040 | 0.080 | 0.064 | 0.043 | 0.021 | 0.043 | - |
| GraphAF | 0.031 | 0.107 | **0.001** | 0.046 | 0.178 | 0.204 | 0.022 | 0.135 | - | - | - | - | - |
| GraphDF | 0.039 | 0.128 | 0.012 | 0.046 | 0.060 | 0.116 | 0.030 | 0.069 | - | - | - | - | - |
| GNF | 0.030 | 0.100 | **0.001** | 0.044 | 0.200 | 0.200 | 0.110 | 0.170 | - | - | - | - | - |
| GraphVAE | 0.137 | 0.166 | 0.051 | 0.118 | 0.358 | 0.969 | 0.551 | 0.626 | 1.594 | **0.000** | 0.904 | 0.833 | - |
| EDP-GNN | 0.054 | 0.092 | 0.007 | 0.051 | 0.050 | 0.159 | 0.027 | 0.079 | 0.460 | 0.243 | 0.316 | 0.340 | 1000 |
| GDSS | 0.027 | **0.033** | 0.008 | 0.022 | 0.044 | 0.098 | 0.009 | 0.058 | 0.133 | 0.009 | 0.123 | 0.088 | 1000 |
| GSDM | - | - | - | - | 0.020 | 0.050 | 0.005 | 0.053 | 0.002 | **0.000** | **0.000** | 0.001 | 1000 |
| DiGress | 0.028 | 0.046 | 0.008 | 0.027 | 0.032 | 0.047 | 0.009 | 0.025 | 0.037 | 0.046 | 0.069 | 0.051 | 500 |
| SwinGNN | 0.017 | 0.060 | 0.003 | 0.027 | **0.006** | 0.125 | 0.018 | 0.050 | **0.000** | **0.000** | **0.000** | **0.000** | 500 |
| GGFlow | **0.005** | **0.033** | 0.004 | **0.014** | 0.011 | **0.030** | **0.002** | **0.014** | 0.030 | **0.000** | 0.016 | 0.015 | 500 |

## 5.2 Molecule Graph Generation

We evaluated GGFlow on two standard molecular datasets, QM9 (Ramakrishnan et al., 2014) and ZINC250k (Irwin et al., 2012), using several metrics: Validity, Validity without correction, Neighborhood Subgraph Pairwise Distance Kernel (NSPDK) Maximum Mean Discrepancy (MMD), and Frechet ChemNet Distance (FCD). To calculate these metrics, we sampled 10,000 molecules. We compared GGFlow against various molecule generation models, including GraphAF, GraphDF, MolFlow (Zang & Wang, 2020), EDP-GNN, GraphEBM (Liu et al., 2021), GDSS, PS-VAE (Kong et al., 2022), MolHF (Zhu et al., 2023), GruM, SwinGNN, DiGress, and GSDM. Detailed descriptions of the datasets, baselines and metrics are provided in Appendix H.

The results, presented in Table 2, indicate that GGFlow effectively captures the distribution of molecular data, showing significant improvements over the baselines. The high Validity without correction suggests that GGFlow successfully learns chemical valency rules. Additionally, GGFlow achieves superior NSPDK and FCD scores on both datasets, demonstrating its ability to generate molecules with distributions closely resembling those of natural molecules. Visualizations of molecules generated by different models are shown in Figure 2, with additional results on GGFlow provided in Appendix J.

Table 2: Generation results on the QM9 and ZINC250k datasets. Results are the means of 3 different runs. The best results and the second-best results are marked **bold** and bold.

| Method | QM9 | | | | ZINC250k | | | | Step |
|---|---|---|---|---|---|---|---|---|---|
| | Val. | Val. w/o corr. | NSPDK | FCD | Val. | Val. w/o corr. | NSPDK | FCD | |
| Training Set | 100 | 100 | 0.0001 | 0.040 | 100 | 100 | 0.0001 | 0.062 | - |
| GraphAF | 100 | 67.14 | 0.0218 | 5.246 | 100 | 67.92 | 0.0432 | 16.128 | - |
| GraphDF | 100 | 83.14 | 0.0647 | 10.451 | 100 | 89.72 | 0.1737 | 33.899 | - |
| MolFlow | 100 | 92.03 | 0.0169 | 4.536 | 100 | 63.76 | 0.0468 | 20.875 | - |
| GraphEBM | 100 | 8.78 | 0.0287 | 6.402 | 100 | 5.29 | 0.2089 | 35.467 | - |
| PS-VAE | - | - | 0.0077 | 1.259 | - | - | 0.0112 | 6.320 | - |
| MolHF | - | - | - | - | 100 | 93.62 | 0.0387 | 23.940 | - |
| EDP-GNN | 100 | 47.69 | 0.0052 | 2.683 | 100 | 83.16 | 0.0483 | 16.819 | 1000 |
| GDSS | 100 | 96.17 | 0.0033 | 2.565 | 100 | 97.12 | 0.0192 | 14.032 | 1000 |
| GSDM | 100 | 99.90 | 0.0034 | 2.614 | 100 | 92.57 | 0.0168 | 12.435 | 1000 |
| GruM | 100 | 99.69 | **0.0002** | 0.108 | 100 | 98.32 | 0.0023 | 2.235 | 1000 |
| SwinGNN | 100 | 99.66 | 0.0003 | 0.118 | 100 | 86.16 | 0.0047 | 4.398 | 500 |
| DiGress | 100 | 98.29 | 0.0003 | **0.095** | 100 | 94.98 | 0.0021 | 3.482 | 500 |
| GGFlow | 100 | **99.91** | **0.0002** | 0.148 | 100 | **99.63** | **0.0010** | **1.455** | 500 |

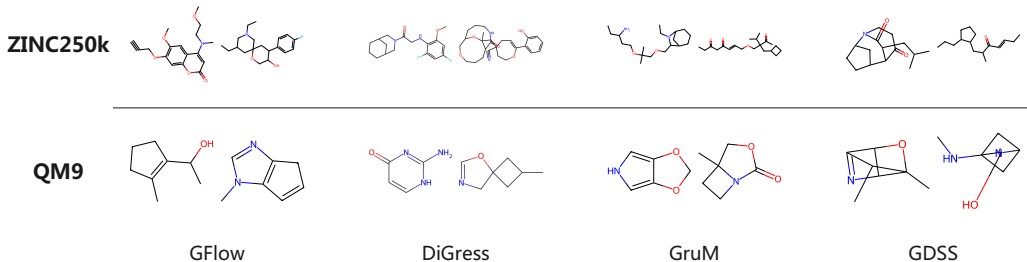

Figure 2: Visualization of generated samples of different models in different molecular datasets

## 5.3 Conditional Generation

To further evaluate the performance of our model, we conducted conditional generation experiments on the QM9 dataset, focusing on generating molecules with molecular properties $\mu$ that closely match a target value $\mu^*$. In the experiment, we set the target value as 1, i.e. $\mu^* = 1$.

For the experiment, we employed a reinforcement learning-based guidance method and compared it to the guided version of DiGress, which also proposes an effective approach for discrete diffusion models in conditional generation tasks. The reward function was defined as $|\mu - \mu^*|$, and the model was trained over 10,000 steps using the training settings detailed in Section 5.2. To evaluate the effectiveness of our guidance method, we compared it against three baselines: (1) Guidance for DiGress (Vignac et al., 2022). (2) Direct supervised training (ST) (3) Supervised fine-tuning (SFT). Additionally, we calculated the mean and variance of $|\mu - \mu^*|$ for samples generated unconditionally by both DiGress and GGFlow to provide a baseline comparison. Further details of the experiment are provided in Appendix H.5.

The results, detailed in Table 3, demonstrate the superiority of our reinforcement learning-based conditional generation method over both ST and SFT approaches. Notably, our method surpasses the guidance techniques used in diffusion models, showcasing its enhanced ability to steer the generative process toward desired outcomes. Additionally, our approach achieves higher validity in conditional generated tasks, highlighting its robustness and superior performance in goal-directed generation.

Table 3: Mean absolute error of molecular property $\mu$ on conditional generation on the QM9 dataset.

| Methods | DiGress | | GGFlow | | | |
|---|---|---|---|---|---|---|
| | Uncondition | +Guidance | Unconditition | Supervised Training | +SFT | +RL |
| Mean | 1.562 | 1.092 | 1.569 | 1.184 | 1.223 | **0.672** |
| Variance | 1.641 | 0.894 | 1.987 | 1.579 | 1.893 | **0.647** |
| Val. w/o corr. | 96.54 | 74.2 | 98.93 | 86.1 | 87.0 | **92.2** |

## 5.4 Ablation studies

To thoroughly assess the individual contributions of the proposed components to GGFlow's performance, we conducted a comprehensive series of ablation experiments. These studies utilized both the Community-small and ZINC250k datasets. Our investigation focused on isolating the impact of: (1) the full GGFlow model, (2) the model performance without the integration of Optimal Transport (OT), denoted as GGFlow (w/o OT), (3) the model performance without the GraphEvo module, denoted as GGFlow (w/o Evo), and (4) the baseline GGFlow framework without either the GraphEvo module or Optimal Transport, designated as GGFlow (w/o both). The specific configurations for each ablated model are summarized in Table 4. The empirical results of these studies are presented in detail in Table 5. Further specifics regarding the experimental setup are provided in Appendix I.2.

Table 4: Ablated model configurations used in the study.

| Method | Flow Matching Framework | Optimal Transport | GraphEvo Module |
|---|---|---|---|
| DiGress | ✗ | ✗ | ✗ |
| GGFlow (w/o both) | ✓ | ✗ | ✗ |
| GGFlow (w/o OT) | ✓ | ✗ | ✓ |
| GGFlow (w/o Evo) | ✓ | ✓ | ✗ |
| GGFlow | ✓ | ✓ | ✓ |

As demonstrated in Table 5, a comparative analysis of GGFlow (w/o OT) and GGFlow reveals that the integration of Optimal Transport consistently improves generation performance. Notably, even in the absence of both GraphEvo and Optimal Transport, GGFlow (w/o both) consistently outperforms DiGress, thereby underscoring the inherent advantages of our flow matching framework over traditional diffusion models in graph generation tasks.

Furthermore, the GraphEvo module, particularly its triangle attention mechanism for edges, effectively captures more intricate node and edge features, leading to substantial performance enhancements. To investigate GraphEvo's ability to extract edge information, we calculated the cross-entropy of edges on the validation set for each epoch in Figure S4. We observed that after incorporating GraphEvo, the edge cross-entropy decreased more rapidly, indicating the model's enhanced capability in capturing edge information.

To evaluate the benefits of Optimal Transport on sampling efficiency, we conducted experiments with varying inference steps, as presented in Figure 3 (More detailed comparison on Community-small datasets in Figure S3). Our findings indicate that GGFlow achieves superior performance compared to other diffusion-based models with significantly fewer inference steps, particularly at 100 and 200 steps. This highlights GGFlow's enhanced sampling efficiency. Furthermore, the integration of Optimal Transport demonstrably boosts sampling efficiency and enhances generative performance across both generic and molecular graph generation tasks, as evidenced by the performance gains observed.

To demonstrate the enhanced training stability conferred by Optimal Transport and the Flow Matching framework, we performed a detailed analysis of evaluation metrics during the training phase. We compared

Table 5: Ablation studies on the OT and GraphEvo on the Commuinty-small and ZINC250k datasets. Results are the means of 3 different runs. The best results are marked **bold**.

| Method | Community-small | | | ZINC250k | | | Step |
|---|---|---|---|---|---|---|---|
| | Deg. | Clus. | Orbit | Val. w/o corr. | NSPDK | FCD | |
| DiGress | 0.032 | 0.047 | 0.009 | 94.98 | 0.0021 | 3.482 | 500 |
| GGFlow (w/o both) | 0.028 | 0.076 | 0.007 | 99.07 | 0.0022 | 2.104 | 500 |
| GGFlow (w/o OT) | 0.018 | **0.027** | 0.004 | 99.58 | 0.0015 | 1.503 | 500 |
| GGFlow (w/o Evo) | 0.015 | 0.055 | 0.005 | 99.12 | 0.0025 | 1.891 | 500 |
| GGFlow | **0.011** | 0.030 | **0.002** | **99.63** | **0.0010** | **1.455** | 500 |

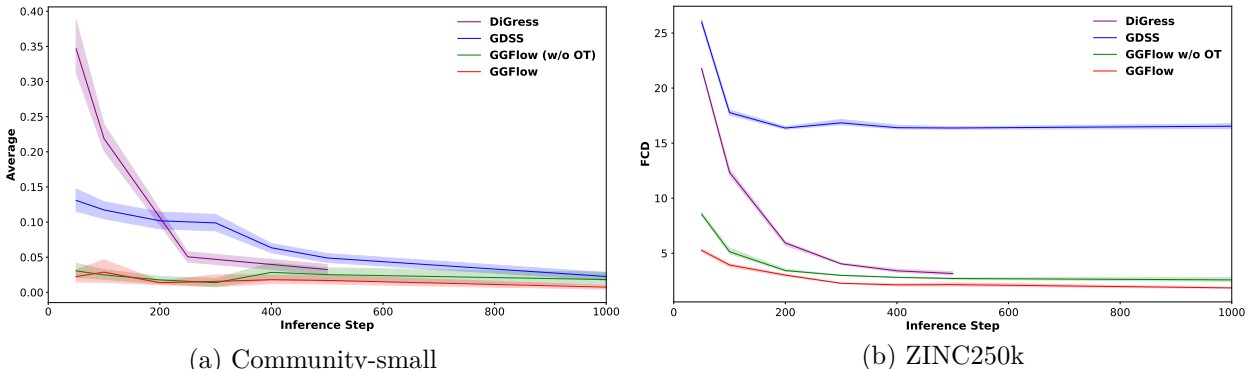

(a) Community-small (b) ZINC250k

Figure 3: Ablation studies on the impact of varying inference steps across Community-small and ZINC250k datasets. Performance of the ZINC250k dataset is evaluated using 1,000 generated molecules due to the computational cost of generating larger sets for metric calculation.

the average values on the Community-small dataset and FCD values on the ZINC250k dataset against DiGress, which shares similar training objectives. For a fair comparison and to highlight the distinct contributions, we specifically focused on GGFlow (w/o both) and GGFlow (w/o Evo). Test set average values and FCD were computed as described in Section 5.1, utilizing checkpoints saved every 100 training steps.

To quantitatively assess training stability, we calculated the Mean Absolute Difference (MAD) of evaluation metric values between adjacent epochs and averaged these differences. As depicted in Figure 4, the presence of Optimal Transport (comparing GGFlow (w/o Evo) to GGFlow (w/o both)) improved model stability, leading to more consistent performance during training.

Figure 4 further illustrates that both GGFlow (w/o both) and GGFlow (w/o Evo) achieve faster and more effective convergence than DiGress. Importantly, GGFlow (w/o Evo) consistently outperforms GGFlow (w/o both) in terms of both average metrics and FCD, clearly demonstrating the benefits derived from Optimal Transport. These findings collectively indicate that both the Optimal Transport module and the underlying Flow Matching framework significantly contribute to greater training stability.

## 6 Conclusion

In this paper, we introduced GGFlow, a discrete flow matching generative model for graphs that incorporates optimal transport and an innovative graph transformer network. GGFlow achieves state-of-the-art performance in unconditional graph generation tasks. Additionally, we presented a novel guidance method using reinforcement learning to control the generative trajectory toward a preferred distribution. Furthermore, our model demonstrates the ability to achieve the best performance across various tasks compared to other baselines, which highlights the practical impact of our guidance method. Despite these advancements, GGFlow has a primary limitation in its scalability to large graphs (with more than 500 nodes). This is primarily

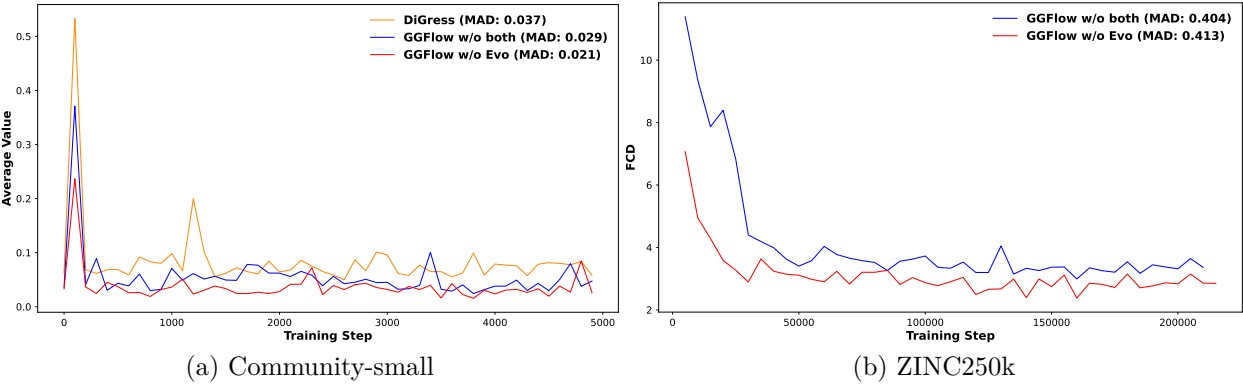

(a) Community-small

(b) ZINC250k

Figure 4: Analysis of training stability for the Flow Matching framework and Optimal Transport on the Community-small and ZINC250k datasets. Test metrics of ZINC250k are calculated using 1,000 generated molecules.

due to the computational complexity of the triangle attention updates and spectral feature computations. Generation times for different graph scales are provided in Appendix I.3. Another area for future work is to improve the theoretical guarantees of our optimal transport method, as its current formulation only provides invariance to identical permutations rather than full permutation equivariance. Future research will focus on enhancing the scalability of GGFlow to larger graphs. We will also explore more generalized and efficient optimal transport methods that are fully permutation-invariant to strengthen the model's theoretical foundation and broaden its applicability.

## 7 Acknowledgments

This work was supported by the National Natural Science Foundation of China under Grants No. 82130055, 32271297, and 32370657; the National Key Research and Development Program of China under Grant No. 2020YFA0907000; the Key Special Project for Technological Innovation and Application Development of Chongqing Municipality under Grant No. CSTB2022TIAD-DEX0035; the King Abdullah University of Science and Technology (KAUST) Office of Research Administration (ORA) under Awards No. REI/1/5289-01-01, REI/1/5992-01-01, URF/1/6713-01-01, and FCC/1/5932-12-11; Center of Excellence for Smart Health (KCSH) under award number 5932; and Center of Excellence on Generative AI under award number 5940. The data underlying this article are available from the first and corresponding author upon reasonable request.

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

## Appendix

## A Background

### A.1 Continuous Flow Matching Generative Model

The generative model aims to establish a mapping $f : \mathbb{R}^d \to \mathbb{R}^d$ that transforms a noise distribution $q_0$ into a target data distribution $q_1$. This transformation is dependent on a density function $p_0$ over $\mathbb{R}^d$, and an integration map $\psi_t$, which induces a pushforward transformation $p_t = [\psi_t]_\#(p_0)$. This denotes the density of points $x \sim p_0$ transported from time 0 to time $t$ along a vector field $u : [0,1] \times \mathbb{R}^d \to \mathbb{R}^d$.

The vector field $u$ is formulated as:

$$\mathrm{d}x = u_t(x)\mathrm{d}t.$$

The solution $\psi_t(x)$ to this ODE, with the initial condition $\psi_0(x) = x$, represents the trajectory of the point $x$ governed by $u$ from time 0 to time $t$.

The evolution of the density $p_t$, viewed as a function $p : [0,1] \times \mathbb{R}^d \to \mathbb{R}$, is encapsulated by the continuity equation:

$$\frac{\partial p}{\partial t} = -\nabla \cdot (p_t u_t),$$

with the initial condition given by $p_0$. Here, $u$ is the probability flow ODE for the path of marginal probabilities $p$, generated over time.

In practical applications, if the probability path $p_t(x)$ and the generating vector field $u_t(x)$ are known and $p_t(x)$ is tractably sampled, we leverage a time-dependent neural network $v_\theta(\cdot, \cdot) : [0,1] \times \mathbb{R}^d \to \mathbb{R}^d$ to approximate $u$. The neural network is trained using the flow matching objective:

$$\mathcal{L}_{\mathrm{FM}}(\theta) = \mathbb{E}_{t \sim \mathcal{U}(0,1), x \sim p_t(x)} \|v_\theta(t,x) - u_t(x)\|^2, \tag{19}$$

which enhances the model's capability to simulate the target dynamics accurately. Avoiding the explicit construction of the intractable vector field, recent works express the probability path as a marginal over a joint involving a latent variable $z$: $p(x_t) = \int p(z)p_{t|z}(x_t|z)$. (Lipman et al., 2022; Tong et al., 2023) and the $p_{t|z}(x_t|z)$ is a conditional probability path, satisfying some boundary conditions at $t = 0$ and $t = 1$.

The conditional probability path also satisfies the transport equation with the conditional vector field $u_t(x|x_1)$:

$$\frac{\partial p_t(x|x_t)}{\partial t} = -\nabla \cdot (u_t(x|x_1)p_t(x_t|x_1)). \tag{20}$$

We can construct the marginal vector field $u_t(x)$ via the conditional probability path $p_{t|1}(x_t|x_1)$ as:

$$u_t(x) = \mathbb{E}_{x_1 \sim p_{1|t}}[u_t(x|x_1)]. \tag{21}$$

We can replace the flow matching loss $\mathcal{L}_{\mathrm{FM}}$ with an equivalent loss regressing the conditional vector field $u_t(x|x_1)$ and marginalizing $x_1$ instead:

$$\mathcal{L}_{\mathrm{CFM}}(\theta) = \mathbb{E}_{\mathcal{U}(t;0,1), x_1 \sim q, x_t \sim p_t(x|x_1)}[u_\theta(t,x) - u_t(x|x_1)].$$
$$\nabla_\theta \mathcal{L}_{\mathrm{FM}}(\theta) = \nabla_\theta \mathcal{L}_{\mathrm{CFM}}(\theta).$$

So we can use $\mathcal{L}_{\mathrm{CFM}}(\theta)$ instead to train the parametric vector field $u_\theta$.

## B Related Works

### B.1 Comparison with Discrete Flow Matching

Campbell et al. (2024) first introduced flow matching in discrete spaces using a continuous-time Markov chain. Building on this, Gat et al. (2024) expanded the framework to encompass general source and target couplings,

including U-coupling and C-coupling. GGFlow advances discrete flow matching and its source-target coupling to suit graph-structured data. Our approach innovatively incorporates efficient optimal transport for graphs within the flow matching framework. To address the inherent sparsity and permutation invariance of graphs, we employ a tailored prior distribution. Additionally, we implement an edge-augmented graph transformer to enhance generative performance and adopt a goal-guided framework for conditional generation. These advancements collectively enhance the practical applicability of GGFlow.

## B.2   Comparison with Graph Discrete Diffusion Model

DiGress (Vignac et al., 2022) and PPGN (Haefeli et al., 2022) were among the first to apply discrete diffusion models to graph generation, highlighting the advantages of discrete state spaces. DiGress further introduced an optimal prior distribution and global structural features specifically designed to enhance graph generation. Their forward and generative processes are expressed as:

$$q(G^t|G^0) = \text{Cat}(G^t, p = G^0\bar{Q}_t), \text{ with } \bar{Q}_t = Q_1 Q_2 \ldots Q_t, \tag{22}$$

$$q(G^{t-1}|G^t, G^0) = \frac{q(G^t|G^{t-1}, G^0)q(G^{t-1}|G^0)}{q(G^t|G^0)} = \text{Cat}(G^{t-1}; p = \frac{G^t Q_t^T \odot G^0 \bar{Q}_{t-1}}{G^0 \bar{Q}_t G^{t*}}), \tag{23}$$

where $G^t$ represents the noisy graph at time $t$, and $Q_t$ is the time-dependent transition matrix. These methods require maintaining convergence properties of the transition matrix and cumulative matrix products, constraining the choice of prior distributions and destabilizing training. In contrast, GGFlow employs a simpler interpolation between the prior and data distributions during training, avoiding cumulative products and improving both training stability and the ease of selecting appropriate priors.

## B.3   Comparison with Graph Discrete Flow Model

GraphDF (Luo et al., 2021) uses a discrete flow model to generate molecular graphs by sequentially sampling discrete latent variables and mapping them to nodes and edges via invertible modulo-shift transforms. GGFlow simplifies this by transforming the invertible modulo-shift into a conditional vector field that interpolates between the prior and data distributions, bypassing the need for complex invertible mappings. Furthermore, while GraphDF adopts an autoregressive process for graph generation, GGFlow generates the entire graph in a one-shot manner, capturing holistic relationships among nodes and edges more efficiently.

## B.4   Comparison with Graph Flow Matching Models

CatFlow (Eijkelboom et al., 2024) employs variational inference to apply flow matching to categorical data, but it only considers the conditional vector field under the assumption of independent coupling in the joint distribution $\pi(G^0, G^1)$ and fails to consider the inherent sparsity of graph structures. and DeFoG (Qin et al., 2024) also applies the discrete flow matching to establish the generation process of the graph, while they don't consider the generalization of the joint distribution. GGFlow extends this by generalizing $\pi(G^0, G^1)$ as an efficient 2-Wasserstein optimal transport map and incorporating an optimal prior distribution tailored for graph structures, improving performance in generation tasks. Additionally, GGFlow introduces a novel goal-directed approach for discrete flow matching in conditional generation tasks, enhancing its practical applicability.

## B.5   Relationship to DiGress

DiGress and GGFlow are both generative models that share similar training objectives, specifically denoiser prediction. In contrast, their mathematical derivations and conceptual starting points differ.

DiGress, a diffusion model, is trained to predict the original data point ($G^0$) from a noisy version ($G^t$). Its training objective, $\mathcal{L}_{\text{DiGress}}$, is derived from the evidence lower bound (ELBO) of the data likelihood, which is gradually loosened to a permutation-invariant loss, as detailed in Lemma 3.2 of Vignac et al. (2022). The objective is defined as:

$$\mathcal{L}_{\mathrm{DiGress}} = \mathbb{E}_{p_{\mathrm{data}}(G^0) q_{t|0}(G^t|G^0)}(\log p_\theta(G|G^t))$$

$$\mathbb{E}_{p_{\mathrm{data}}(G^0)}(-\log p_\theta(G^0)) \leq \mathcal{L}_{\mathrm{DT}} \leq \sum_{k=1}^{T} \mathbb{E}_{p_{\mathrm{data}}(G^0) q_{k|0}(G^k|G^0)}(-\log p_\theta(G|G^k)).$$

$$\mathcal{L}_{\mathrm{DT}}(\theta) = \mathbb{E}_{p_{\mathrm{data}}(G_0)} \left[ \mathrm{KL}(q_{K|0}(G^K|G^0)||p_{\mathrm{ref}}(G^K)) - \mathbb{E}_{q_{1|0}(G^1|G^0)} \left[ \log p_\theta^{0|1}(G^0|G^1) \right] \right.$$
$$\left. + \sum_{k=1}^{K-1} \mathbb{E}_{q_{k+1|0}(G^{k+1}|G^0)} \left[ \mathrm{KL}(q_{k|k+1,0}(G^k|G^{k+1},G^0)||p_{k|k+1}^\theta(G^k|G^{k+1})) \right] \right]$$

The full derivation of the training objective and inequality, detailed in Proposition 9 in Campbell et al. (2022), shows how the ELBO is manipulated to arrive at this loss function.

GGFlow, a flow-matching model, also uses a denoiser prediction objective, but its derivation is rooted in different principles. The training objective, $\mathcal{L}_{\mathrm{GGFlow}}$, is derived from an ELBO of the flow-matching process, as detailed in Section C.2 of Campbell et al. (2024):

$$\mathcal{L}_{\mathrm{GGFlow}} = \mathbb{E}_{p_{\mathrm{data}}(G^1) \mathcal{U}(t;0,1) \pi(G^0,G^1) p_t(G^t|G^0,G^1)}[\log \hat{p}_{1|t}(G^1|G^t, G^0)]$$
$$\mathbb{E}_{p_{\mathrm{data}}(G^1)}(-\log p_\theta(G^1)) \leq -\mathcal{L}_{\mathrm{ELBO}} = -(\mathcal{L}_{\mathrm{GGFlow}} + \mathcal{L}_R + \mathcal{L}_{\mathrm{KL}} + C) \leq -(\mathcal{L}_{\mathrm{GGFlow}} + \mathcal{L}_R + C).$$
$$\mathcal{L}_{\mathrm{R}} = \mathbb{E}_{\mathcal{U}(t;0,1) p_t(G^t)}[-R_t^\theta(G^t) + \mathbb{E}_{p_{1|t}(G^1|G^t)}[R_t(G^t|G^1)] \log R_t^\theta(G^t)]$$

Unlike DiGress, the GGFlow ELBO includes an additional term, $\mathcal{L}_R$. As argued in Section C.2 of Campbell et al. (2024), this term is often deemed unnecessary as it encourages the generative rate to match a chosen jump rate, simplifying the final training objective to just $\mathcal{L}_{\mathrm{GGFlow}}$. Fortunately, this simplified training objective also satisfies permutation invariance, a shared desirable property with DiGress.

## C Proofs

### C.1 Optimal Prior Distribution

This prior is structured as a product of a single distribution $v$ for all nodes and a single distribution $e$ for all edges, $\prod_i v \times \prod_{i,j} e$, to ensure exchangeability across the graph components.

**Theorem 2** (Optimal prior distribution). *Consider the class $\mathcal{C} = \{\prod_i u \times \prod_{i,j} v, (u,v) \in \mathcal{P}(\mathcal{V}) \times \mathcal{P}(\mathcal{E})\}$ of distributions over graphs, which factorize as the product of a uniform distribution $v$ over node attribute space $\mathcal{V}$ and a uniform distribution $e$ over edge attribute space $\mathcal{E}$. Given any arbitrary distribution $P$ over graphs (viewed as a tensor of order $n + n^2$), with $q_V$ and $q_E$ as its marginal distributions for node and edge attributes respectively, then the orthogonal projection of $P$ onto $\mathcal{C}$ is defined as $\phi^G = \prod_i q_V \times \prod_{i,j} q_E$. This projection minimizes the Euclidean distance:*

$$\phi^G \in \arg \min_{(v,e) \in \mathcal{C}} \|P - \prod_{1 \leq i \leq n} v \times \prod_{1 \leq i,j \leq n} e\|_2^2.$$

The details and proof of Theorem 2 are extensively discussed in DiGress (Vignac et al., 2022).

### C.2 Choice of conditional velocity field

The subsequent theoretical discussion is structured into three parts, each establishing a fundamental result necessary for our proposed framework:

1. Lemma 1 demonstrates that the vector field $u_t(G^t, G)$, defined as an expectation over the conditional vector field $u_t(G^t, G|G^0, G^1)$ (Equation 11), satisfies the Kolmogorov forward equation, provided the conditional rate adheres to the conditional Kolmogorov equation. This finding establishes the fundamental consistency of the vector field and extends Proposition 3.1 in Campbell et al. (2024).

2. We then introduce a specific functional form for the conditional vector field. Proposition 3 proves that this proposed formulation satisfies the conditional Kolmogorov equation. This result is an extension of Proposition 3.2 found in Campbell et al. (2024).

3. Finally, for practical implementation and computational efficiency, Corollary 1 presents a simplified, implementable expression for the conditional vector field, derived directly from Proposition 3.

The specific points of departure between our proof techniques and the original proofs by Campbell et al. (2024) are indicated using blue text for ease of reference and comparison.

**Lemma 1.** *If $u_t(G^t, G|G^0, G^1) \triangleq \prod_{x \in \mathcal{V} \cup \mathcal{E}} u_t(x^t, x|x^0, x^1)$ is a conditional vector field that satisfies the conditional Kolmogorov equation*

$$\partial_t p_{t|1}(x^t|x^1, x^0) = \sum_{x \neq x^t} u_t(x^t, x|x^0, x^1) p_{t|1}(x|x^1, x^0) - \sum_{x \neq x^t} u_t(x^t, x|x^0, x^1) p_{t|1}(x^t|x^1, x^0), \qquad (24)$$

*for each single generic random variable $x \in \mathcal{V} \cup \mathcal{E}$, then the vector field*

$$u_t(G, G^t) \triangleq \mathbb{E}_{p_{1|t}(G^1, G^0|G)}(u_t(G^t, G|G^0, G^1)) = \prod_{x \in \mathcal{V} \cup \mathcal{E}} \mathbb{E}_{p_{1|t}(x^1, x^0|x)}(u_t(x^t, x|x^0, x^1)) \qquad (25)$$

*also satisfies the Kolmogorov equation $\partial_t p_t(x^t) = \sum_{x \neq x^t} u_t(x^t, x) p_{t|1}(x) - \sum_{x \neq x^t} u_t(x^t, x) p_{t|1}(x^t)$.*

*Proof.* We assume the graph generation process can be modeled by a time-inhomogeneous Markov Chain, where the component-wise transition is defined by the conditional vector field $u_t(x^t, x|x^0, x^1)$. Our goal is to derive the marginal vector field $u_t(x^t, x)$ and confirm that the marginal distribution $p_t(x^t)$ satisfies the Kolmogorov forward equation. We use the fact $u_t(x^t, x^t|x^0, x^1) = -\sum_{x \neq x_t} u_t(x^t, x|x^0, x^1)$ for compactness. We denote the joint distribution of the initial state $G^0$ and the terminal state $G^1$ as $\pi(G^0, G^1)$, which implies $\pi(x^0, x^1)$ for a generic variable $x \in \mathcal{V} \cup \mathcal{E}$.

$$\partial_t p_{t|1}(x^t|x^1, x^0) = \sum_x u_t(x^t, x|x^0, x^1) p_{t|1}(x|x^1, x^0)$$

$$\mathbb{E}_{\pi(x^1, x^0)}[\partial_t p_{t|1}(x^t|x^1, x^0)] = \mathbb{E}_{\pi(x^1, x^0)}[\sum_x u_t(x^t, x|x^0, x^1) p_{t|1}(x|x^1, x^0)]$$

$$\partial_t \mathbb{E}_{\pi(x^1, x^0)}[p_{t|1}(x^t|x^1, x^0)] = \sum_x \sum_{(x^0, x^1)} \pi(x^0, x^1) u_t(x^t, x|x^0, x^1) p_{t|1}(x|x^1, x^0)$$

Using the identity $\pi(x^0, x^1) p_{t|1}(x|x^1, x^0) = p_t(x) p_{1|t}(x^1, x^0|x)$ (derived from Bayes' Rule where $p_{1|t}(x^1, x^0|x)$ is the posterior distribution of the endpoints given the intermediate state $x$):

$$\partial_t p_t(x^t) = \sum_G \sum_{(x^0, x^1)} p_{1|t}(x^1, x^0|x) u_t(x^t, x|x^0, x^1) p_t(x) \quad \text{based on Bayes Rule}$$

$$\partial_t p_t(x^t) = \sum_x [\sum_{(x^0, x^1)} p_{1|t}(x^1, x^0|x) u_t(x^t, x|x^0, x^1)] p_t(x)$$

$$\partial_t p_t(x^t) = \sum_x u_t(x^t, x) p_t(x)$$

The intermediate steps demonstrate the specific algebraic transformation required for our modified coupling, which represents a key point of departure from Campbell et al. (2024). Since the overall graph transition is assumed to be component-wise independent, the vector field $u_t(G^t, G)$ also satisfies the Kolmogorov forward equation. □

**Proposition 3.** *We then prove that the proposed conditional vector field,*

$$u_t(G^t, G|G^0, G^1) = \prod_{x \in \mathcal{V} \cup \mathcal{E}} u_t(x^t, x|x^0, x^1) = \prod_{x \in \mathcal{V} \cup \mathcal{E}} \frac{ReLU(\partial_t p_{t|1}(x|x^1, x^0) - \partial_t p_{t|1}(x^t|x^1, x^0))}{\mathbf{Z}_t \cdot p_{t|1}(x^t|x^1, x^0)} \qquad (26)$$

*satisfies the conditional Kolmogorov equation. where* $\text{ReLU}(a) = \max(a, 0)$ *and* $\mathbf{Z}_t = |\{x^t : p_t(x^t|x^0, x^1) > 0\}|$. $u_t(x^t, x|x^0, x^1) = 0$ *when* $p_t(x|x^1, x^0) = 0$ *and* $p_t(x^t|x^1, x^0) = 0$. *When* $x^t = x$, *the conditional vector field* $u_t(x^t, x^t|x^0, x^1) = -\sum_{x^t \neq x} u_t(x^t, x|x^0, x^1)$. *We assume that if* $p_{t|1}(x|x^1, x^0) = 0$, *then* $\partial_t p_{t|1}(x|x^1, x^0) = 0$.

*Proof.* The proof proceeds by verifying that the conditional Kolmogorov equation is satisfied by the conditional vector field $u_t(x^t, x|x^0, x^1)$ for each single generic random variable $x \in \mathcal{V} \cup \mathcal{E}$.

This conditional vector field, inspired by discrete flow matching, is defined as:

$$u_t(x^t, x|x^0, x^1) = \frac{\text{ReLU}(\partial_t p_{t|1}(x^t|x^1, x^0) - \partial_t p_{t|1}(x|x^1, x^0))}{\mathbf{Z}_t \cdot p_{t|1}(x|x^1, x^0)}$$

where $\mathbf{Z}_t = |\{x^t : p_t(x^t|x^0, x^1) > 0\}|$ is a normalization constant. We first consider the case where $p_{t|1}(x^t|x^1, x^0) > 0$. the Kolmogorov Equation requires: $\mathbf{LHS} = \partial_t p_{t|1}(x^t|x^1, x^0)$. The Right-Hand Side (RHS) is defined by the flux balance:

$$
\begin{aligned}
\mathbf{RHS} = & \sum_{x \neq x^t, p_{t|1}(x|x^1,x^0)>0} \frac{\text{ReLU}(\partial_t p_{t|1}(x^t|x^1, x^0) - \partial_t p_{t|1}(x|x^1, x^0))}{\mathbf{Z}_t \cdot p_{t|1}(x|x^1, x^0)} p_{t|1}(x|x^1, x^0) \\
& - \sum_{x \neq x^t, p_{t|1}(x|x^1,x^0)>0} \frac{\text{ReLU}(\partial_t p_{t|1}(x|x^1, x^0) - \partial_t p_{t|1}(x^t|x^1, x^0))}{\mathbf{Z}_t \cdot p_{t|1}(x^t|x^1, x^0)} p_{t|1}(x^t|x^1, x^0) \\
= & \frac{1}{\mathbf{Z}_t} \sum_{x \neq x^t, p_{t|1}(x|x^1,x^0)>0} \text{ReLU}(\partial_t p_{t|1}(x^t|x^1, x^0) - \partial_t p_{t|1}(x|x^1, x^0)) \\
& - \frac{1}{\mathbf{Z}_t} \sum_{x \neq x^t, p_{t|1}(x|x^1,x^0)>0} \text{ReLU}(\partial_t p_{t|1}(x|x^1, x^0) - \partial_t p_{t|1}(x^t|x^1, x^0)) \\
= & \frac{1}{\mathbf{Z}_t} \sum_{x \neq x^t, p_{t|1}(x|x^1,x^0)>0} (\partial_t p_{t|1}(x^t|x^1, x^0) - \partial_t p_{t|1}(x|x^1, x^0)) \\
= & \frac{\mathbf{Z}_t - 1}{\mathbf{Z}_t} \partial_t p_{t|1}(x^t|x^1, x^0) - \frac{1}{\mathbf{Z}_t} \sum_{x \neq x^t, p_{t|1}(x|x^1,x^0)>0} \partial_t p_{t|1}(x|x^1, x^0) \\
= & \frac{\mathbf{Z}_t - 1}{\mathbf{Z}_t} \partial_t p_{t|1}(x^t|x^1, x^0) - \frac{1}{\mathbf{Z}_t} \partial_t (1 - p_{t|1}(x^t|x^1, x^0)) \\
= & \frac{\mathbf{Z}_t - 1}{\mathbf{Z}_t} \partial_t p_{t|1}(x^t|x^1, x^0) + \frac{1}{\mathbf{Z}_t} \partial_t p_{t|1}(x^t|x^1, x^0) \\
= & \partial_t p_{t|1}(x^t|x^1, x^0) = \mathbf{LHS}
\end{aligned}
$$

In the case that $p_{t|1}(x^t|x^1, x^0) = 0$ by assumption we have that $\partial_t p_{t|1}(x^t|x^1, x^0) = 0$. We have both $u_t(x^t, x|x^0, x^1) = 0$ and $u_t(x, x^t|x^0, x^1) = 0$ because $p_{t|1}(x^t|x^1, x^0) = 0$. Therefore we have $\mathbf{LHS} = \mathbf{RHS} = 0$ and thus the Kolmogorov equation is satisfied

$\square$

**Corollary 1.** *We can simplify the conditional vector field in the Proposition 3 as:*

$$
\begin{aligned}
u_t(G^t, G|G^0, G^1) = & \prod_{x \in \mathcal{V} \cup \mathcal{E}} u_t(x^t, x|x^0, x^1) = \prod_{x \in \mathcal{V} \cup \mathcal{E}} \frac{ReLU(\partial_t p_{t|1}(x|x^1, x^0) - \partial_t p_{t|1}(x^t|x^1, x^0))}{\mathbf{Z}_t \cdot p_{t|1}(x^t|x^1, x^0)} \\
= & \prod_{x \in \mathcal{V} \cup \mathcal{E}} \left( \frac{\delta\{x^t, x^0\}}{\mathbf{Z}_t(1-t)} \left(1 + \delta\{x, x^1\} - \delta\{x, x^0\}\right) \right),
\end{aligned}
$$

*where* $\text{ReLU}(a) = \max(a, 0)$ *and* $\mathbf{Z}_t = |\{x : p_t(x|x^0, x^1) > 0, x \in \mathcal{V} \cup \mathcal{E}\}|$. $u_t(x^t, x|x^0, x^1) = 0$ *when* $p_{t|1}(x^t|x^1, x^0) = 0$. *When* $x^t = x$, *the conditional vector field* $u_t(x^t, x^t|x^0, x^1) = -\sum_{x^t \neq x} u_t(x^t, x|x^0, x^1)$.

*Proof.* Equation 7 establishes the factorization $p_t(G^t) = \prod_i p(v_i^t) \prod_{j,k} p(e_{jk}^t)$. This implies that the conditional probability path and the resulting vector field decouple over the graph components. We therefore derive the rate matrix for a single generic random variable $x \in \mathcal{V} \cup \mathcal{E}$ (representing any specific node or edge).

Consider the conditional probability path $p_{t|1}(x) = t\delta\{x, x^1\} + (1-t)\delta\{x, x^0\}$. We define the time-derivative vector field $v(x) \triangleq \partial_t p_{t|1}(x)$. Differentiating with respect to $t$ yields:

$$v(x) = \delta\{x, x^1\} - \delta\{x, x^0\} = \begin{cases} 1 & \text{if } x = x^1, \\ -1 & \text{if } x = x^0, \\ 0 & \text{otherwise.} \end{cases} \tag{27}$$

We define the numerator of the rate equation as $N(x^t, x) = \text{ReLU}(v(x) - v(x^t))$. We analyze $N$ based on the value of the current state $x^t$. Since the probability path $p_{t|1}(x)$ concentrates all probability mass on the interpolation between noise $x^0$ and data $x^1$, we restrict our analysis to these two supported cases:

- **Case I:** $x^t = x^1$. Here $v(x^t) = 1$. Since $v(x) \leq 1$, $N = 0$.

- **Case II:** $x^t = x^0$. Here $v(x^t) = -1$. The term is $\text{ReLU}(v(x) + 1)$.
  - If moving to target $(x = x^1)$: $N = 2$.
  - If staying $(x = x^0)$: $N = 0$.
  - If moving to noise $(x \neq x^0, x^1)$: $N = 1$.

  This is expressed as: $N = 1 + \delta\{x, x^1\} - \delta\{x, x^0\}$.

Finally, we consider the denominator $D(x^t) = \mathbf{Z}_t \cdot p_{t|1}(x^t | x^1, x^0)$. The flow is non-trivial only in Case II, where $p_{t|1}(x^t | x^1, x^0) = 1 - t$. Combining the numerator from Case II with this denominator yields the component-wise rate:

$$u_t(x^t, x | x^0, x^1) = \frac{\delta\{x^t, x^0\}}{\mathbf{Z}_t(1-t)} \left(1 + \delta\{x, x^1\} - \delta\{x, x^0\}\right). \tag{28}$$

Since the noise structure factorizes over the graph components (nodes and edges) as shown in Equation 7, the generation process is equivalent to solving independent flow matching problems for each component. Consequently, the vector field for the entire graph $G$, denoted as $u_t(G^t, G | G^0, G^1)$, is not a scalar field but a collection of independent component-wise rates.

We define the global rate tensor $u_t(G^t, G | G^0, G^1)$ element-wise for all elements $x$ in the sets $\mathcal{V}$ and $\mathcal{E}$:

$$u_t(G^t, G | G^0, G^1) \triangleq \prod_{x \in \mathcal{V} \cup \mathcal{E}} u_t(x^t, x | x^0, x^1) = \prod_{x \in \mathcal{V} \cup \mathcal{E}} \left( \frac{\delta\{x^t, x^0\}}{\mathbf{Z}_t(1-t)} \left(1 + \delta\{x, x^1\} - \delta\{x, x^0\}\right) \right). \tag{29}$$

$\square$

### C.3 Proof of Proposition 1

*Proof.* The Kolmogorov forward equations for discrete flow matching are expressed as:

$$\partial_t p_t = u_t p_t, \tag{30}$$

If we establish the permutation invariance of the prior distributions $p_{\text{ref}}$ and the permutation equivariance of conditional flow probabilities, then it follows that $p(G^1)$ is permutation exchangeable.

According to Theorem 2, we deduce the permutation invariance of the prior distribution $p_{\text{ref}}$. Given the conditional probabilities $p(G^{t+\Delta t} | G^t) = \delta\{G^{t+\Delta t}, G^t\} + \hat{u}_t(G^t, G^{t+\Delta t})\Delta t$, it suffices to demonstrate the

permutation equivariance of the conditional probabilities. This requires showing the permutation equivariance of the vector field $u_t$. Consider the case for nodes:

$$\sigma u_t^V(V_i^t, V_i^{t+\Delta t}) = \sigma \left( \mathbb{E}_{\hat{p}_{1|t}^V(V_i^1|V_i^t)}[u_t^V(V_i^t, V_i^{t+\Delta t}|V_i^1, V_i^0)] \right),$$

$$\text{LHS} = u_t^V(V_{\sigma^{-1}(i)}^t, V_{\sigma^{-1}(i)}^{t+\Delta t}),$$

$$\text{RHS} = \left( \mathbb{E}_{\hat{p}_{1|t}^V(V_{\sigma^{-1}(i)}^1|V_{\sigma^{-1}(i)}^t)}[u_t^V(V_{\sigma^{-1}(i)}^t, V_{\sigma^{-1}(i)}^{t+\Delta t}|V_{\sigma^{-1}(i)}^1, V_{\sigma^{-1}(i)}^0)] \right),$$

$$= u_t^V(V_{\sigma^{-1}(i)}^t, V_{\sigma^{-1}(i)}^{t+\Delta t}) = \text{LHS}.$$

where $\sigma$ is a permutation operator. This establishes the permutation equivariance of $u_t$ and the exchangeability of the generated distribution. □

## C.4 Proof of Theorem 1

First, we want to clarify the rationale and foundation of our theorem. The goal of optimal transport is to pair source and target data points with minimal cost during training, which is beneficial for our interpolation (Bose et al., 2023; Song et al., 2024). Thus, we design our optimal transport approach from the perspective of interpolation.

We define the node order of the graph $G$ as the order of the nodes and edges in matrix representation. For example, if the node set of $G$ is $\{A, B, C\}$, the possible node orders include $(A, B, C)$, $(B, A, C)$ or $(C, B, A)$.

In the interpolation process, we transform the graph representation to a matrix representation before performing interpolation. For example, for source data $G^0 = (V^0, E^0), V^0 \in \mathbb{R}^{a \times n}, E^0 \in \mathbb{R}^{a \times a \times m}$ and target data $G^1 = (V^1, E^1), V^1 \in \mathbb{R}^{a \times n}, E^1 \in \mathbb{R}^{a \times a \times m}$, where $a$ is the number of nodes, $n$ is the class number of nodes, and $m$ is the class number of edges, the node orders of $G^0$ and $G^1$ have been fixed. Therefore, interpolation is performed directly on these fixed node orders.

The optimal transport aims to find pairs with the minimum cost for interpolation, and the interpolation is conducted on a fixed node order. Additionally, during optimal transport calculations, we also utilize the matrix representation of these graphs and our prior distribution is permutation invariant. Therefore, we aim to match source data with the target data $G^1$ whose node order is fixed, to achieve minimal transport cost. Furthermore, we assume that all pairs of source and target data share the same node order during optimal transport, which also facilitates the identification of pairs with minimal cost.

Regarding the permutation of the intermediate graph $G^t$, we have $\sigma G^t = t\sigma G^0 + (1-t)\sigma G^1$, where $G^0$ and $G^1$ share an identical permutation. Our network $\hat{p}^1(G^1|G^0, G^t)$ needs to maintain permutation equivariance, such that $\hat{p}^1(G^1|\sigma G^0, \sigma G^t) = \sigma \hat{p}^1(G^1|G^0, G^t)$ for any permutation $\sigma$ to approximate $\sigma G^1$. So we prove the invariance of optimal transport under identical permutations, i.e. $\pi(G^0, G^1) = \pi(\sigma G^0, \sigma G^1)$.

*Proof.* Building on the foundations established in Theorem 2 and Proposition 1, we confirm the permutation invariance of both the target and source distributions. The Hamming distance is invariance under identical permutations $\pi$, as shown by:

$$H(G^0, G^1) = \sum_i \delta(v_i^0, v_i^1) + \frac{1}{2} \sum_{i,j} \delta(e_{ij}^0, e_{ij}^1)$$

$$= \sum_i \delta(v_{\sigma^{-1}(i)}^0, v_{\sigma^{-1}(i)}^1) + \frac{1}{2} \sum_{i,j} \delta(e_{\sigma^{-1}(i)\sigma^{-1}(j)}^0, e_{\sigma^{-1}(i)\sigma^{-1}(j)}^1)$$

$$= H(\sigma G^0, \sigma G^1)$$

This property of the Hamming distance ensures the invariance of the optimal transport map $\pi$ under identical permutations. □

Additionally, the prior distribution is permutation invariant and our GraphEvo is permutation equivariant, all permutations of graphs are generated with equal probability (Eijkelboom et al., 2024).

**Lemma 2.** *Let $p_0(G)$ be an exchangeable distribution and our model $\hat{p}_{1|t}(G^1|G^t, G^0)$ is permutation equivariant. Then, all permutations of the generated graphs are selected with equal probability.*

*Proof.* As the permutation equivariance of our model $\hat{p}_{1|t}(G^1|G^t, G^0)$, implies the equivariance of our vector fields $u_t$. Moreover, the sampling procedure exhibits permutation equivariance, where $\pi$ is a permutation.

$$\hat{p}_{1|t}(G^1|\sigma G^t, \sigma G^0) = \sigma \hat{p}_{1|t}(G^1|G^t, G^0), \quad t = 0$$
$$p_{\Delta t|0}(G^{\Delta t}|\sigma \hat{G}^1, \sigma G^0) = \delta\{G^{\Delta t}, \sigma G^0\} + u_0(G^{\Delta t}, \sigma G^0|\sigma G^0, \sigma \hat{G}^1)\Delta t$$
$$= \sigma[\delta\{G^{\Delta t}, G^0\} + u_0(G^{\Delta t}, G^0|G^0, \hat{G}^1)\Delta t] = \sigma p_{\Delta t|0}(G^{\Delta t}|\hat{G}^1, G^0), \quad t = 0$$
$$p_{t+\Delta t|t}(G^{\Delta t+t}|\sigma \hat{G}^1, \sigma G^t, \sigma G^0) = \delta\{G^{\Delta t+t}, \sigma G^t\} + u_t(G^{\Delta t+t}, \sigma G^t|\sigma G^0, \sigma \hat{G}^1)\Delta t$$
$$= \sigma[\delta\{G^{\Delta t+t}, G^t\} + u_t(G^{\Delta t+t}, G^t|G^0, \hat{G}^1)\Delta t] = \sigma p_{t+\Delta t|t}(G^{\Delta t+t}|\hat{G}^1, G^t, G^0), \quad t = \Delta t, \ldots, 1 - \Delta t$$

Therefore, since $p_0$ assigns equal density to all permutations of $G$, the resulting distribution $p_1$ preserves this property. $\square$

## D    Details of GraphEvo

GraphEvo is a novel edge-augmented graph transformer model designed for graph data. To enhance the generative capabilities of GGFlow, GraphEvo introduces a triangle update mechanism, which significantly improves the exchange of edge information. We incorporate FiLM and PNA layers into our architecture (Vignac et al., 2022):

$$\text{FiLM}(X_1, X_2) = X_1(\text{Linear}(X_2) + 1) + \text{Linear}'(X_2)$$
$$\text{PNA}(X) = \text{Linear}\Big(\text{Cat}(\max(X), \min(X), \text{mean}(X), \text{std}(X))\Big).$$

The time complexity of GraphEvo is $O(N^3)$.

GraphEvo integrates global structural features to improve generation performance, including both graph-theoretic and domain-specific attributes:

**Graph-theoretic features**: These encompass node-level properties such as the number of $k$-cycles ($k \leq 5$) containing this point and an estimate of the largest connected component, alongside graph-level metrics like the total number of $k$-cycles ($k \leq 6$) and connected components.

**Molecular features**: These account for the current valency of each atom and the molecular weight of the entire molecule.

### D.1    Proof of Proposition 2

*Proof.* Let $G^t = (V^t, E^t)$ is a intermediate graph, and $\sigma G^t = (\sigma^* V, \sigma^* E \sigma)$ is the permutation. To prove the permutation properties of the graph, we need to consider two aspects: additional structural features and the model architecture.

First, the spectral and structural features are permutation equivariant for node-level features and invariant for graph-level features. Additionally, the FiLM blocks and Linear layers are permutation equivariant, while the PNA pooling function is permutation invariant. Layer normalization is also permutation equivariant.

As GraphEvo is built using permutation equivariant components, we conclude that the overall model is permutation equivariant.

$\square$

# E    Analysis of Optimal Transport

To analyze the time complexity of optimal transport (OT), we compared the training time of OT with that of DiGress, using identical architectures on an NVIDIA A100 80G GPU. We evaluated the effects of model size, batch size, and number of nodes by measuring the duration of single training steps across three different datasets. Our results indicate that the time required for OT accounts for only 5% of the total training time, highlighting the efficiency of our optimal transport.

Table S1: Time Complexity of Optimal Transport

| Dataset | Planar | Zinc250k | Community-small |
|---|---|---|---|
| DiGress Training Time (s) | 0.1647 | 0.1690 | 0.0456 |
| GGFlow Training Time (s) | 0.1264 | 0.1301 | 0.0408 |
| Optimal Transport Time (s) | 0.0025 | 0.0070 | 0.0024 |
| Percentage of OT | 1.9% | 5.3% | 5.6% |
| Model Size (M) | 3.6 | 4.6 | 6.4 |
| Batch Size | 64 | 128 | 80 |
| Number of Nodes | 64 | [6,38] | [12,20] |

To further demonstrate the advantages of our proposed distance metric, we conducted a comparative analysis using Fused-Gromov-Wasserstein (FGW), which is defined in Vayer et al. (2018), as an alternative permutation-invariant graph distance for model training.

A primary challenge encountered with FGW was its computational overhead. We observed that calculating the FGW distance on both the Community-small and ZINC250k datasets required several orders of magnitude more time than our model's typical training duration. As detailed in Table S2, the computation time for FGW on ZINC250k was particularly prohibitive, preventing us from completing a full training run for a comparative model. Consequently, a comprehensive model trained with FGW was only feasible on the Community-small dataset.

Table S2: Time comparison of Optimal Transport distance computation

| Dataset | Zinc250k | Community-small |
|---|---|---|
| Hamming Distance Training Time (s) | 0.0070 | 0.0024 |
| Fused Gromov-wasserstein Distance Training Time (s) | 60.3062 | 9.0301 |

Table S3 presents the generative performance comparison on the Community-small dataset. Our model, utilizing the Hamming distance, outperformed the FGW-based model on the Cluster and Orbit metrics, while exhibiting a slightly reduced performance on the Degree metric.

Table S3: Generative results with different Optimal Transport distances on Community-small dataset

| Method | Degree | Cluster | Orbit |
|---|---|---|---|
| Hamming Distance | 0.011 | 0.030 | 0.002 |
| Fused Gromov-wasserstein Distance | 0.004 | 0.069 | 0.006 |

More critically, our model demonstrated superior training stability compared to the FGW-based model, as evidenced by the validation KL divergence on the validation set. Figure S1 illustrates this enhanced stability. This improved stability is directly attributable to our distance metric being specifically tailored and better adapted to the Flow Matching training methodology.

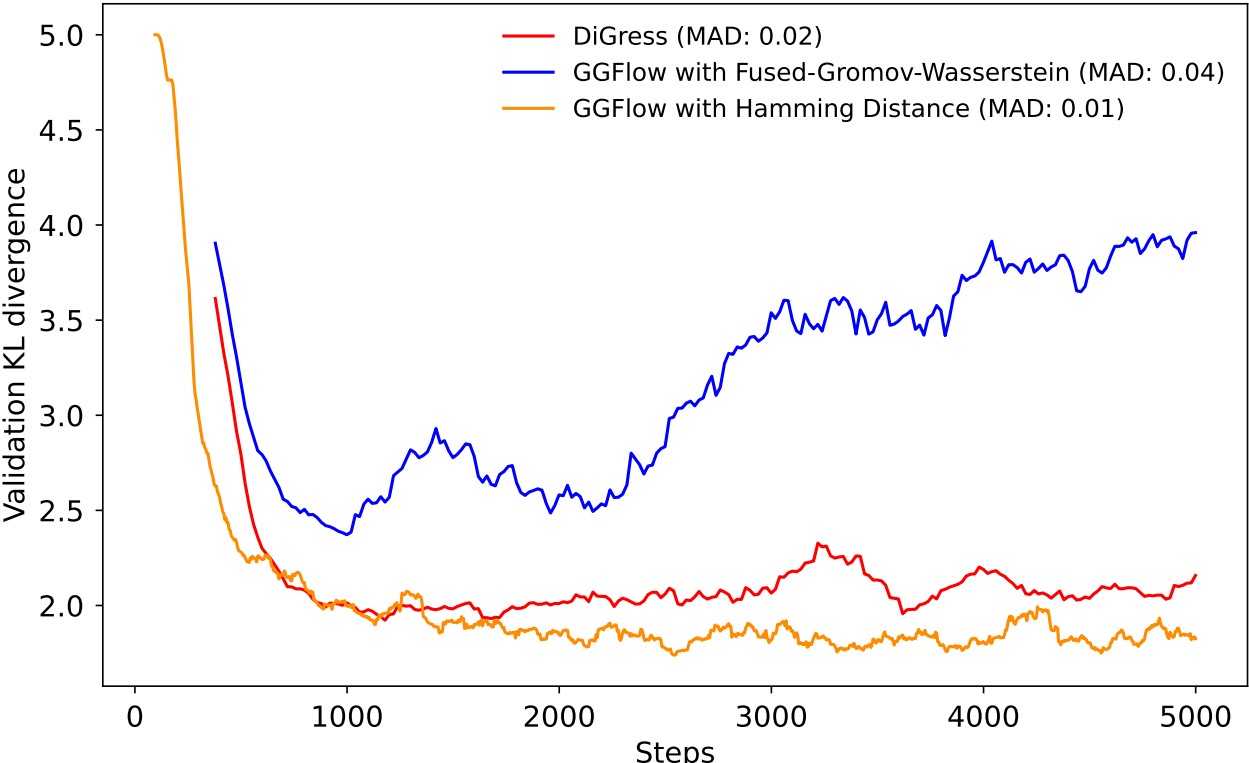

Figure S1: Validation KL Divergence comparison of different Optimal Transport approaches on Community-small dataset

These findings collectively indicate that, while our proposed optimal transport approach limits invariance to identical permutations, it does not negatively impact the model's practical performance and is better suited and more efficient for our Flow Matching model than Fused-Gromov-Wasserstein.

## F Toy example of goal-guided graph generation

We demonstrate the utility of our goal-guided framework of flow matching with a toy example, depicted in Figure S2: (a) shows a trained unconditional flow matching model mapping noise distribution $p_0$ to data distribution $p_1$. (b, c) illustrate the effect of temperature $T$ on the exploration, with higher temperatures resulting in broader data point distribution. (d) shows how fine-tuning according to Equation 18 concentrates data in regions with higher rewards. (e-f) illustrate the corresponding flow matching trajectories.

## G Additional Experiments Results

In this section, we present additional metrics including Spectre (Spec.) and Validity&Novelty&Uniquess (Val.&Nov.&Uni.) across general graph datasets including the Planar and Enzymes datasets, as summarized in Tables S4, S5 and S6. The MMD kernel in the planar dataset followed the GruM (Jo et al., 2024). We also include the standard deviation of our results in Table S7, illustrating the consistency and superior performance of our method.

To further compare GGFlow with baseline models, we measured the MMD between the test datasets and a set of 1,024 generated graphs in the Ego-small and Community-small datasets. The results in Table S8 demonstrate that GGFlow achieves the highest performance across all metrics, significantly outperforming other baseline models.

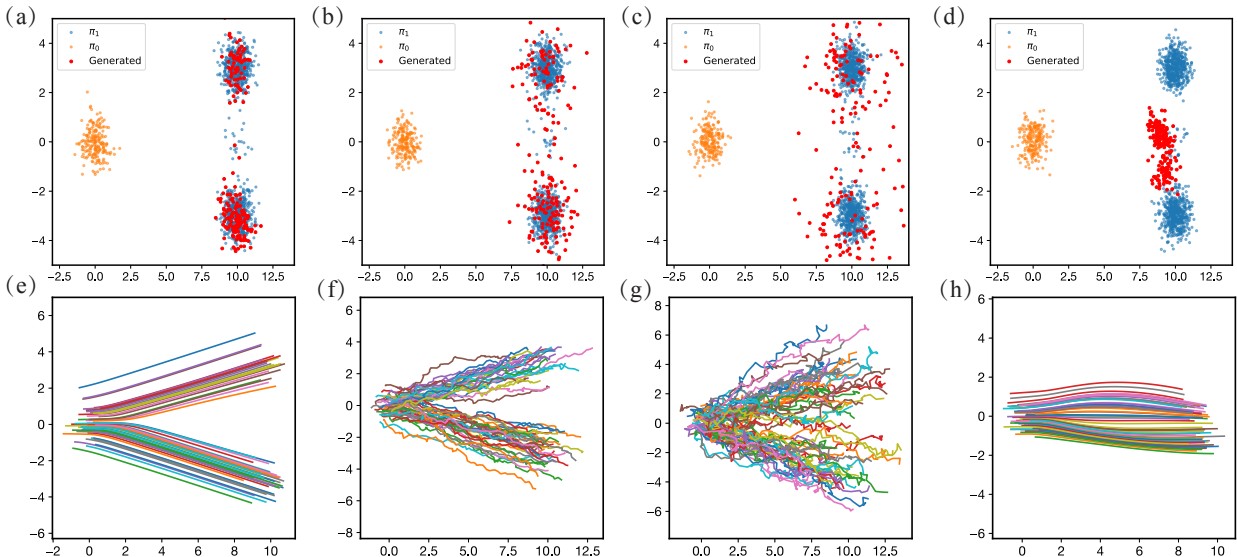

Figure S2: (a-d) Data distribution of the flow matching model, $\pi_0$ is the original distribution (orange), $\pi_1$ is the target data distribution (blue), and the red dots are the data distribution generated by the model. (e-h) In reinforcement learning, the flow matching model conducts exploration/sampling trajectories

Table S4: Additional generation results on the generic graph datasets. Results are the means of 3 different runs. The best results are marked **bold**.

| Method | Ego-small | | Community-small | | Grid | | Step |
|---|---|---|---|---|---|---|---|
| | Spec. | Nov.&Uni. | Spec. | Nov.&Uni. | Spec. | Nov.&Uni. | |
| Training Set | 0.006 | 30 | 0.012 | 100 | 0.009 | 25 | - |
| GDSS | 0.034 | 27.5 | 0.053 | 100.0 | 0.043 | 100.0 | 1000 |
| GSDM | - | - | **0.024** | 0.0 | 0.015 | 0.0 | 1000 |
| DiGress | 0.017 | 30.0 | 0.055 | 100.0 | 0.025 | 100.0 | 500 |
| SwinGNN | 0.016 | **52.5** | 0.025 | 55.0 | **0.008** | 100.0 | 500 |
| GGFlow | **0.006** | 32.5 | 0.031 | **100.0** | 0.022 | **100.0** | 500 |

Table S5: Generation results on the planar graph datasets. The best results are marked **bold**. $\sigma$ denotes the standard deviation.

| Method | Planar | | | | | Step |
|---|---|---|---|---|---|---|
| | Deg. | Clus. | Orbit | Spec. | Val.&Nov.&Uni. | |
| Training Set | 0.0002 | 0.0165 | 0.0002 | 0.0050 | 100 | - |
| GDSS | 0.0039 | 0.2593 | 0.1732 | 0.0370 | 0.0 | 1000 |
| GRASP | 0.0022 | 0.2749 | 0.0055 | 0.0098 | 0.0 | 200 |
| DiGress | **0.0003** | 0.0372 | 0.0098 | 0.0106 | 75.0 | 500 |
| GruM | 0.0004 | 0.0382 | 0.0095 | **0.0069** | 87.5 | 1000 |
| GGFlow | 0.0156 | **0.0196** | **0.0019** | 0.0091 | **97.5** | 500 |
| $\sigma$ | 0.0064 | 0.0037 | 0.0006 | 0.0012 | 2.5 | - |

Table S6: Generation results on the Enzymes graph datasets. The best results are marked **bold**. $\sigma$ denotes the standard deviation.

| Method | Enzymes | | | | Step |
|---|---|---|---|---|---|
| | Deg. | Clus. | Orbit | Avg. | |
| Training Set | 0.008 | 0.096 | 0.012 | 0.039 | - |
| GraphRNN | 0.017 | 0.043 | 0.021 | 0.043 | - |
| GraphAF | 1.669 | 1.283 | 0.266 | 1.073 | - |
| GraphDF | 1.503 | 1.061 | 0.202 | 0.922 | - |
| GraphVAE | 1.369 | 0.629 | 0.191 | 0.730 | - |
| EDP-GNN | 0.023 | 0.268 | 0.082 | 0.124 | 1000 |
| GDSS | 0.026 | 0.102 | 0.009 | 0.046 | 1000 |
| GSDM | 0.013 | 0.088 | 0.010 | 0.037 | 1000 |
| DiGress | 0.010 | 0.046 | 0.002 | 0.019 | 500 |
| GGFlow | **0.008** | **0.026** | **0.002** | **0.012** | 500 |
| $\sigma$ | 0.0041 | 0.0106 | 0.0008 | 0.0130 | - |

Table S7: Standard deviation and mean of generation results on the general graph datasets. $\mu$ and $\sigma$ denote the mean and standard deviation, respectively

| Metric | Ego-small | | | | Community-small | | | | Grid | | | |
|---|---|---|---|---|---|---|---|---|---|---|---|---|
| | Deg. | Clus. | Orbit | Sepc. | Deg. | Clus. | Orbit | Sepc. | Deg. | Clus. | Orbit | Sepc. |
| $\mu$ | 0.005 | 0.033 | 0.005 | 0.008 | 0.011 | 0.030 | 0.002 | 0.031 | 0.030 | 0.000 | 0.016 | 0.022 |
| $\sigma$ | 0.007 | 0.012 | 0.003 | 0.001 | 0.006 | 0.012 | 0.002 | 0.002 | 0.008 | 0.000 | 0.003 | 0.001 |

Table S8: Generation results on the generic graph datasets with 1024 generated graphs. The best results are marked **bold**.

| Method | Ego-small | | | | Community-small | | | | Step |
|---|---|---|---|---|---|---|---|---|---|
| | Deg. | Clus. | Orbit | Sepc. | Deg. | Clus. | Orbit | Spec. | |
| GraphRNN | 0.040 | 0.050 | 0.060 | - | 0.030 | 0.010 | 0.010 | - | - |
| GNF | 0.010 | 0.030 | 0.001 | - | 0.120 | 0.150 | 0.200 | - | - |
| EDP-GNN | 0.010 | 0.025 | 0.003 | - | 0.006 | 0.127 | 0.018 | - | 1000 |
| GDSS | 0.023 | 0.020 | 0.005 | 0.047 | 0.029 | 0.068 | 0.004 | 0.151 | 1000 |
| GSDM | - | - | - | - | **0.003** | 0.008 | 0.0009 | 0.011 | 1000 |
| DiGress | 0.017 | 0.038 | 0.006 | 0.021 | 0.013 | 0.040 | 0.004 | 0.055 | 500 |
| SwinGNN | 0.004 | 0.023 | 0.003 | 0.023 | 0.003 | 0.088 | 0.010 | 0.016 | 500 |
| GGFlow | **0.004** | **0.004** | **0.0008** | **0.009** | 0.004 | **0.003** | **0.0006** | 0.018 | 500 |

# H   Implementation Details

## H.1   Algorithms of GGFlow

Details of the training procedure and guided training procedure are provided in Algorithm 3 and 4.

## H.2   Baselines Implementation

To benchmark the performance of GGFlow, we ensure consistency by using identical splits of training and test sets across all datasets. Below, we provide the implementation details for each baseline model. To guarantee

---
**Algorithm 3** Training Procedure of GGFlow
---
**Require:** Training set $S_{\text{train}} = \{G = (V, E)\}, p_{\text{ref}},$
 1: **for** $n \in \{0, \dots, N_{\text{iter}} - 1\}$ **do**
 2:     $t \in \mathcal{U}(0, 1)$, Sample $G^1$ from $S_{\text{train}}$
 3:     $G^0 = (V^0, E^0) \sim p_{\text{ref}}$
 4:     $(G^0, G^1) \sim \pi(G^0, G^1)$
 5:     // Sample from conditional probability flow.
 6:     $V^t \sim (t\delta\{V^t, V^1\} + (1 - t)\delta\{V^t, V^0\})$ and $E^t \sim (t\delta\{E^t, E^1\} + (1 - t)\delta\{E^t, E^0\})$
 7:     $\hat{p}_{1|t}(G^1|G^t, G^0) = \text{GraphEvo}_{\theta_n}(G^t, t)$
 8:     $\mathcal{L} = \mathbb{E}_{p_{\text{data}}(G^1)\mathcal{U}(t;0,1)\pi(G^0,G^1)p_t(G^t|G^0,G^1)}[\log \hat{p}_{1|t}(G^1|G^t, G^0)]$
 9:     $\theta_{n+1} = \text{optimizer\_update}(\theta_n, \mathcal{L})$
10: **end for**
11: $\theta^* = \theta_{N_{\text{iter}}}$
12: **return** $\theta^*$

---
**Algorithm 4** Training Procedure of Guided GGFlow by Reinforcement Learning
---
**Require:** $\theta_0, \theta, \alpha, \beta, \tau, N_{\text{steps}},$ traj, $G^0 \sim p_{\text{ref}}, T, N_{\text{train}}$
 1: $\theta \leftarrow \theta_0$
 2: **for** $i \in \{1, \dots, N_{\text{train}}\}$ **do**
 3:     $\Delta t = 1/N_{\text{steps}}$
 4:     Collect flow trajectory $\left(G^0, t = 0, \mathcal{R}(G^0, 0)\right)$ in traj.
 5:     **for** $n \in \{0, \dots, N_{\text{steps}} - 1\}$ **do**
 6:         $\hat{p}_{1|t}(G^1|G^t, G^0), = \text{GraphEvo}(G^t, t)$
 7:         Get $G^{t+\Delta t}$ by sampling from Equation 17.
 8:         $(V^{t+\Delta t}, E^{t+\Delta t}) = G^{t+\Delta t}$
 9:         $t = t + \Delta t$
10:         Compute the reward function $\mathcal{R}(G^{t+\Delta t}, t + \Delta t)$.
11:         Collect flow trajectory $\left(G^{t+\Delta t}, t + \Delta t, \mathcal{R}(G^{t+\Delta t}, t + \Delta t)\right)$ in traj.
12:     **end for**
13:     Update network using Equation 18.
14:     $t = 0$
15: **end for**
16: **return** Guided flow matching model $\theta^*$

---

a fair comparison, most baseline models are retrained three times, and the average results from these runs are reported as the final outcomes in unconditional generation tasks. The results of the DeepGMG, GraphRNN and GNF for Ego-small and Community-small dataset are taken from their original papers.

**GraphAF** (Shi et al., 2019) We follow the implementation guidelines provided in the TorchDrug tutorials (`https://torchdrug.ai/docs/tutorials/generation.html`).

**GraphDF** (Shi et al., 2019) Model scripts are sourced from the DiG repository (`https://github.com/divelab/DIG/tree/dig-stable/examples/ggraph/GraphDF`).

**GraphVAE** (Shi et al., 2019) Scripts are obtained from the GraphVAE section of the GraphRNN repository (`https://github.com/JiaxuanYou/graph-generation/tree/master/baselines/graphvae`).

**MoFlow** (Zang & Wang, 2020) Implementation scripts are taken from the MoFlow repository (`https://github.com/calvin-zcx/moflow`).

**GraphEBM** (Liu et al., 2021) We use the implementation available in the GraphEBM repository (`https://github.com/biomed-AI/GraphEBM`).

**EDP-GNN** (Niu et al., 2020)  The model is implemented according to the scripts in the EDP-GNN repository (`https://github.com/ermongroup/GraphScoreMatching`).

**GDSS** (Jo et al., 2022b)  Implementation details are sourced from the GDSS repository (`https://github.com/harryjo97/GDSS`).

**GSDM** (Luo et al., 2023)  Scripts are implemented from the GSDM repository (`https://github.com/ltz0120/Fast_Graph_Generation_via_Spectral_Diffusion`).

**PS-VAE** (Kong et al., 2022)  Implementation details are sourced from the PS-VAE repository (`https://github.com/THUNLP-MT/PS-VAE`).

**MolHF** (Zhu et al., 2023)  The model is implemented according to the scripts in the MolHF repository (`https://github.com/violet-sto/MolHF`).

**GRASP** (Minello et al., 2024)  Implementation details are sourced from the GRASP repository (`https://github.com/lcosmo/GRASP`).

**SwinGNN** (Yan et al., 2023)  Implementation details are sourced from the SwinGNN repository (`https://github.com/DSL-Lab/SwinGNN`). The authors employ the 'gaussian_tv' MMD kernel, whereas other methods use 'gaussian_emd' or 'gaussian'. To ensure a fair comparison, we adopt the same kernel.

**GruM** (Jo et al., 2024)  Scripts are implemented from the GruM repository (`https://github.com/harryjo97/GruM/`).

**DiGress** (Vignac et al., 2022)  The implementation is based on the DiGress repository (`https://github.com/cvignac/DiGress`).

### H.3  Details of Generic Datasets

#### H.3.1  Dataset

**Ego-small**  This dataset consists of 200 small one-hop ego graphs derived from the Citeseer network (Sen et al., 2008). Each graph contains between 4 and 18 nodes.

**Community-small**  This dataset includes 100 random community graphs, each formed by two communities of equal size generated using the E-R model (Erdős et al., 1960) with a probability parameter of $p = 0.7$. The graphs range in size from 12 to 20 nodes.

**Enzymes**  The dataset comprises 587 protein graphs, with each graph representing the tertiary structure of enzymes sourced from the BRENDA database (Schomburg et al., 2004), which have between 10 and 125 nodes.

**Grid**  The dataset consists of 100 standard 2D grid graphs with $100 \leq |V| \leq 400$.

**Planar**  The dataset consists of 200 planar graphs, each with 64 nodes, generated using Delaunay triangulation on uniformly distributed random points.

Table S9: Statistics of the generic graph datasets

| Dataset | type | Number of graphs | Number of nodes |
|---|---|---|---|
| Ego-small | Real | 200 | [4, 18] |
| Community-small | Synthetic | 100 | [12, 20] |
| Enzymes | Real | 587 | [10, 125] |
| Planar | Synthetic | 200 | 64 |
| Grid | Synthetic | 100 | [100,400] |

### H.4 Details of Molecule Datasets

#### H.4.1 Dataset

**QM9** It is a subset of the GDB-17 database and consists of 134,000 stable organic molecules, each containing up to 9 heavy atoms: carbon, oxygen, nitrogen, and fluorine (Ramakrishnan et al., 2014). The dataset includes 12 tasks related to quantum properties. We follow the train/test split from GDSS, using 12,000 molecules for training and the remaining 1,000 for testing.

**ZINC250k** It contains 250,000 drug-like molecules with a maximum of 38 atoms per molecule (Irwin et al., 2012). It includes 9 atom types and 3 edge types. For a fair comparison, we use the same train/test split as previous works, such as GDSS and GSDM.

Table S10: Statistics of the molecular graph datasets

| Dataset | type | Number of graphs | Number of nodes | Number of node types | Number of edge types |
|---------|------|------------------|-----------------|----------------------|----------------------|
| QM9 | Real | 133,885 | [1, 9] | 4 | 3 |
| ZINC250k | Real | 249,455 | [6, 38] | 9 | 3 |

#### H.4.2 Metrics

For generic graph datasets, we employ Maximum Mean Discrepancy (MMD) to assess the distributions of graph statistics, specifically degree distribution, clustering coefficient, the number of occurrences of 4-node orbits, and eigenvalues of the normalized graph Laplacian. In alignment with prior research (Jo et al., 2022b), we utilize specialized kernels for MMD calculations: the Gaussian Earth Mover's Distance (EMD) kernel for degree distribution and clustering coefficient, the Gaussian Total Variation (TV) kernel for eigenvalues of the normalized graph Laplacian, and a standard Gaussian kernel for the 4-node orbits. To ensure a fair comparison, the size of the prediction set matches that of the test set.

**Validity** We permit atoms to exhibit formal charges during valency checks because of the presence of formal charges in the training molecules. It is the fraction of valid molecules after valency correction or edge resampling.

**Validity w/o correction** This metric explicitly evaluates the quality of molecule generation before any correction phase, providing a baseline for raw generation performance.

**FCD** FCD quantifies the functional connectivity density within a molecule by computing distances and connectivity between atoms, based on both structural and chemical features. It describes the three-dimensional structure, topological features, and chemical properties of molecules, making it valuable in fields such as drug design, compound screening, and molecular simulations.

**NSPDK** NSPDK assesses molecular similarity by comparing shortest paths within their graphical structures. It captures connectivity patterns and chemical environments, effectively describing relationships and similarities between molecules. For two distributions $p$ and $q$, the MMD using NSPDK is calculated as:

$$\text{MMD}^2_{\text{NSPDK}}(p,q) = \frac{1}{n(n-1)} \sum_{i=1}^{n} \sum_{j\neq i}^{n} k_{\text{NSPDK}}(\mathcal{X}_i, \mathcal{X}_j) + \frac{1}{m(m-1)} \sum_{i=1}^{m} \sum_{j\neq i}^{m} k_{\text{NSPDK}}(\mathcal{Y}_i, \mathcal{Y}_j) \tag{31}$$

$$- \frac{2}{mn} \sum_{i=1}^{n} \sum_{j=1}^{m} k_{\text{NSPDK}}(\mathcal{X}_i, \mathcal{Y}_j) \tag{32}$$

Here, $k_{\text{NSPDK}}(\cdot)$ denotes the NSPDK kernel function. $\mathcal{X}$ is the set of molecules from distribution $p$. $\mathcal{Y}$ is the set of molecules from distribution $q$. $n$ and $m$ represent the number of samples drawn from distributions $p$ and $q$, respectively. This formula quantifies the difference between the distributions $p$ and $q$ using the NSPDK kernel.

### H.5 Details of Conditional Generation

We included three guidance baselines in our conditional generation task:

**DiGress model with guidance** Utilizing the guidance method integrated into the DiGress model (Vignac et al., 2022).

**Direct supervised training (ST)** It involved selecting training samples from the dataset that satisfied $|\mu - \mu^*| < 1.0$ and retraining them using supervised learning settings identical to those in Section 5.2.

**Supervised fine-tuning (SFT)** This method involved fine-tuning a pre-trained GGFlow model on molecules generated with $|\mu - \mu^*| < 1.0$, maintaining the same training settings as in Section 5.2.

These models were trained over 10,000 steps using the training settings detailed in Section 5.2. We then generated 1,000 samples to calculate the results for each guidance method and the unconditional method, with the values of $\mu$ estimated using Psi4 (Smith et al., 2020). We set the hyperparameters $\alpha$ and $\beta$ as 0.999 and 0.001.

## I Experiment Settings

### I.1 Hyperparameter Settings

Table S11 presents the hyperparameters employed in our experimental setup. For each dataset, the final resutls in Table 1 and Table 2 are the means of 5 different runs.

Table S11: Hyperparameter settings of different datasets

| Hyperparameter | Ego-small | Community-small | Grid | Planar | Enzymes | QM9 | ZINC250k |
|---|---|---|---|---|---|---|---|
| Number of layers | 5 | 7 | 5 | 4 | 6 | 9 | 9 |
| Hidden dimension of X | 256 | 256 | 256 | 256 | 256 | 256 | 128 |
| Hidden dimension of E | 128 | 128 | 128 | 128 | 128 | 128 | 64 |
| Hidden dimension of y | 128 | 128 | 128 | 128 | 128 | 128 | 64 |
| Optimizer | Adamw | Adamw | Adamw | Adamw | Adamw | Adamw | Adamw |
| Learning rate | $2 \times 10^{-4}$ | $2 \times 10^{-4}$ | $2 \times 10^{-4}$ | $2 \times 10^{-4}$ | $2 \times 10^{-4}$ | $2 \times 10^{-4}$ | $2 \times 10^{-4}$ |
| Batch size | 64 | 128 | 4 | 64 | 8 | 512 | 128 |
| Number of epochs | 2000 | 3000 | 5000 | 5000 | 10000 | 1000 | 1000 |
| Number of sampling steps | 500 | 500 | 500 | 500 | 500 | 500 | 500 |

### I.2 Ablation Studies Settings

For the evaluation of varying inference steps, we followed the same experimental settings as outlined in Sections 5.1 and 5.2. Samples were generated for 10 runs. The results were then visualized using the mean and variance across these 10 runs. It is important to note that in DiGress, the number of inference steps is constrained by its predefined diffusion steps ($N = 500$), so the DiGress curve terminates at 500 inference steps.

For the ablation studies of GGFlow without Optimal Transport (GGFlow w/o OT), GGFlow without GraphEvo (GGFlow w/o Evo) and GGFlow without GraphEvo and optimal transport (GGFlow w/o both), we adhered to the settings described in Sections 5.1 and 5.2. The final results were obtained by averaging the outcomes from five different runs.

Due to the excessive time required to compute metrics for 10,000 generated molecules, we conducted the experiments on inference steps and model stability using a smaller set of 1,000 generated molecules to calculate the test metrics on ZINC250k dataset.

To further investigate the advantages of optimal transport, we present generation results with varying inference steps on the Community-small dataset. As shown in Figure S3, GGFlow demonstrates superior generation quality compared to GGFlow (w/o OT), exhibiting narrower confidence intervals and comparable performance

with fewer inference steps, which suggests that optimal transport enhances sampling both efficiency and stability.

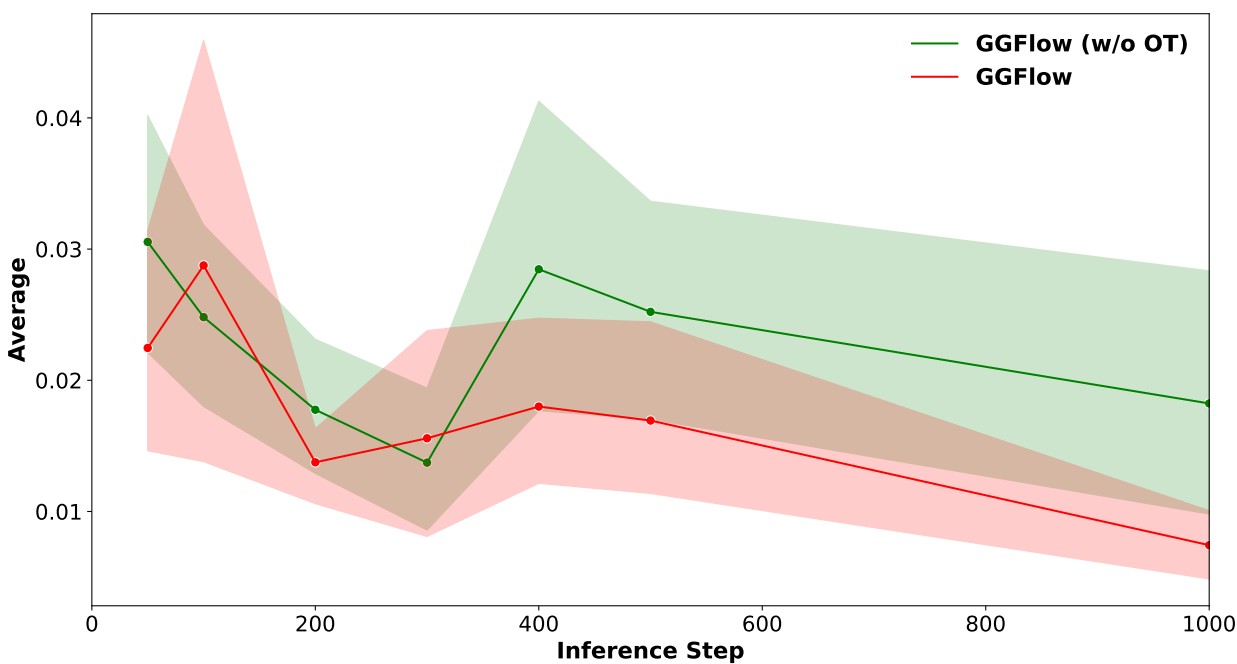

Figure S3: Ablation studies of varying inference steps on Community-small dataset

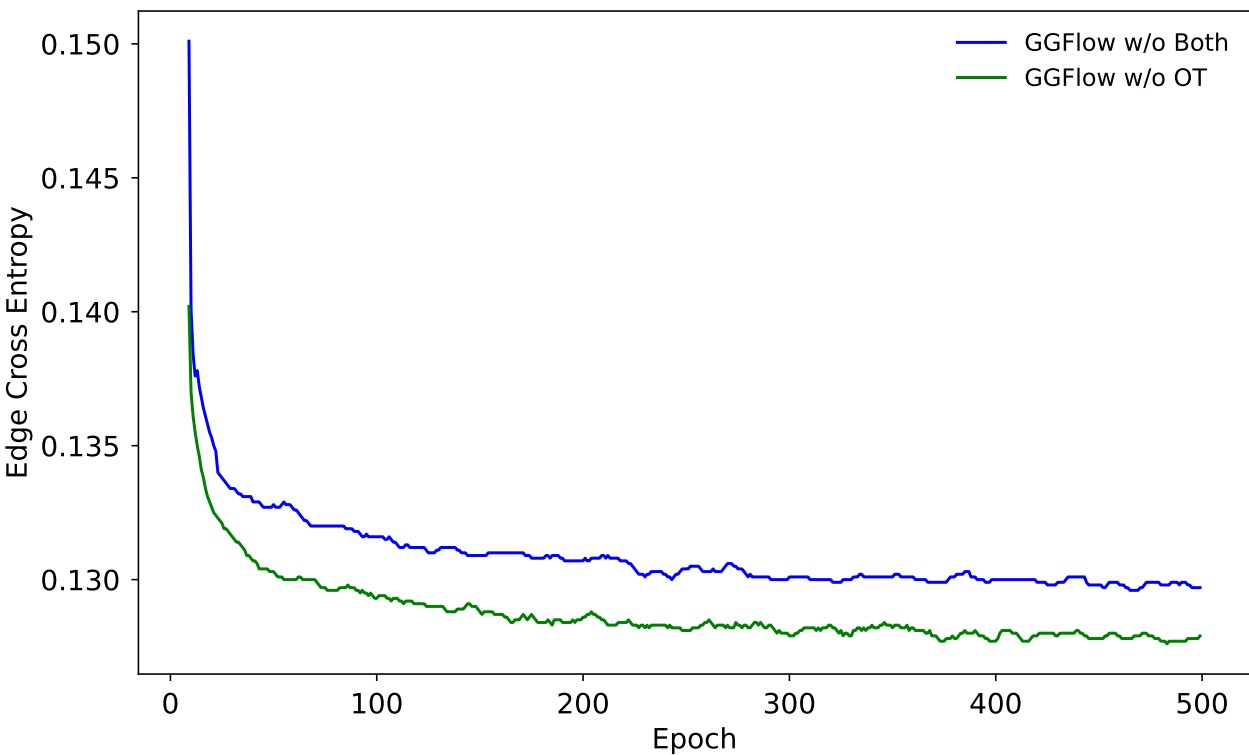

Figure S4: Cross-entropy of edges on the validation set for each epoch on ZINC250k dataset

### I.3 Generation Times with Different Graph Scales

We conducted experiments to evaluate generation times across various graph scales using the Enzymes dataset. The GraphEvo model was configured with six layers, and generation time was measured for a single graph on an NVIDIA A100 80G GPU.

Table S12: Generation times with different graph scales.

| Number of Nodes | 10 | 50 | 100 | 200 | 400 |
|:---:|:---:|:---:|:---:|:---:|:---:|
| Time (s) | 1.92 | 4.24 | 12.15 | 48.59 | 235.8 |

## J   Visualization

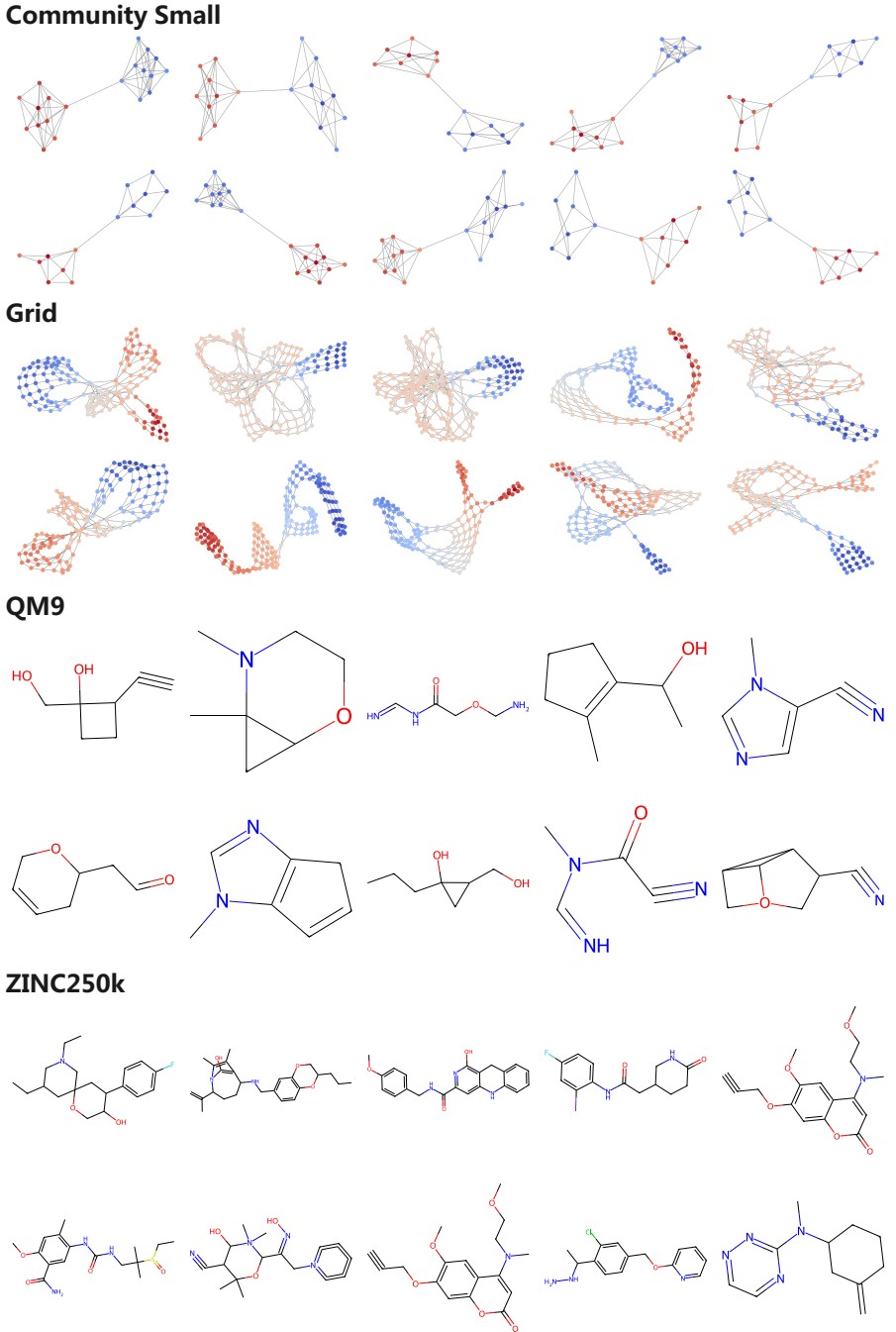

Figure S5: Visualization of generated samples of our model in different datasets

