# OpenReview forum: "GGFlow: A Graph Flow Matching Method with Efficient Optimal Transport"
_TMLR — Accepted by TMLR_

### Review · Reviewer_duAs · 2025-05-13

**Summary Of Contributions:**

The paper develops an approach called GGFlow which is a discrete flow matching generative model for graphs. To improve generative performance the authors augments the approach with optimal transport and an edge a-augmented graph transformer as well as a goal guided framework for conditional generation. Empirical evidence is provided for improved generation over the baselines.

**Audience:**

Yes

**Broader Impact Concerns:**

No concerns

**Claims And Evidence:**

No

**Requested Changes:**

1. Please explain the design decisions behind each component. Specifically explain how optimal transport is adapted for graph generation, why is it efficient and how does it aid to training stability and sampling efficiency.  As the work is developed on existing discrete flow matching models, please motivate/ elaborate on any non-trivial adaptations that are made to make specific components work for graphs.

 2. I found another related work which also seems to focus on sampling inefficiency as the current work : https://arxiv.org/abs/2410.04263. I would request the authors to add a discussion on how the current work is different from DeFoG.

**Strengths And Weaknesses:**

**Strengths**
1. The paper introduces new modules to improve discrete flow matching for graph generation. The approach shows empirical improvements over existing approaches.

**Weaknesses**
 While a number of components have been introduced there is a lack of discussion on the design decisions and the actual contributions of the paper. For example, the addition of optimal transport to discrete flow matching model is stated as the first contribution but the claimed efficient optimal transport is explained in a. short paragraph in section 3.2. It is not clear what exactly is done to make it efficient for graphs and how does it lead to improvement in sampling efficiency and training stability.
Similarly the design decisions for the other two components are missing.

---

> ### Author Response · Authors · 2025-06-01
> **Explanation of Design Decisions and Contributions of Each Component**
>
> We sincerely thank the reviewer for their insightful comments and constructive feedback
>
> ## Explanation of Design Decisions and Contributions of Each Component
>
> ### Efficient Optimal Transport
> The objective of optimal transport (OT) methods in our context is to pair data points with initial noise samples during training such that the transport cost is minimized. This allows the model to encounter noise samples that are "closest" to the data during training, which in turn boosts generation efficiency, as the model can produce plausible samples closest to the initial noise during the generation process.
>
> Defining the transport cost is crucial in optimal transport. In graph computation, determining the edit distance between two graphs is an NP-hard problem. Traditional methods often require extensive computation time, which is prohibitive during model training. To address this, we designed a corresponding transport cost—**Graph Hamming Distance**—specifically for our Flow Matching model. We detail the rationale behind this design in Appendix C.4. This method avoids the need for graph traversal for edit distance calculation, thus dramatically improving computational efficiency.
>
> #### Experiments:
> we have conducted an in-depth comparative analysis using Fused-Gromov-Wasserstein (FGW), which is defined in Equation 1 of Vayer et.al (2018) [1], a widely recognized permutation-invariant graph distance, as an alternative for training our model. Our findings are summarized below and detailed in Appendix E of the revised manuscript.
>
> **Computational Efficiency**: A primary challenge encountered with FGW was its significant computational overhead. As shown in Table 1, calculating the FGW distance on both the Community-small and ZINC250k datasets required several orders of magnitude more time than our model's typical training duration. Specifically, the computation time for FGW on ZINC250k was prohibitive, preventing us from completing a full training run for a comparative model. Consequently, a comprehensive model trained with FGW was only feasible on the Community-small dataset.
>
> | Datasets | community small | zinc250k |
> | --- | --- | --- |
> | Hamming Distance Time | 0.0024 | 0.0070 |
> | Fused Gromov-wasserstein Distance Time | 9.0301 | 60.3062 |
>
> **Generative Performance and Training Stability**: As presented in Table 2, our model, utilizing the Hamming distance, outperformed the FGW-based model on the Cluster and Orbit metrics, while exhibiting a slightly reduced performance on the Degree metric on the Community-small dataset.
>
> |  | Degree | Cluster | Orbit |
> | --- | --- | --- | --- |
> | Hamming Distance OT | 0.011 | 0.030 | 0.002 |
> | Fused Gromov-wasserstein Distance Time | 0.004 | 0.069 | 0.006 |
>
> More critically, our model demonstrated higher training stability compared to the FGW-based model, as evidenced by the validation KL divergence on the validation set. Figure S1 in the Appendix illustrates this enhanced stability with a more consistent and lower KL divergence trajectory. This improved stability is directly attributable to our distance metric being specifically tailored and better adapted to the Flow Matching training methodology, which inherently benefits from a simpler and more efficient distance for interpolating trajectories.
>
> These findings collectively indicate that, while our proposed optimal transport approach limits invariance to identical permutations (i.e., it is sensitive to node ordering, but the flow matching framework learns to align this), it does not negatively impact the model's practical performance and is substantially more efficient and better suited for our Flow Matching model than Fused-Gromov-Wasserstein for the task of discrete graph generation. The efficiency gain allows for scaling to larger datasets like ZINC250k, where FGW becomes computationally prohibitive.
>
>
> [1] Vayer, Titouan, et.al, Optimal transport for structured data with application on graphs, arXiv, 2018

---

> ### Author Response · Authors · 2025-06-01
> **Explanation of Design Decisions and Contributions of Each Component**
>
> ## Explanation of Design Decisions and Contributions of Each Component
>
> ### GraphEvo Module
> The design of the GraphEvo framework addresses a common limitation in many existing graph transformer architectures: their primary focus on node-centric attention, often at the expense of adequately capturing complex edge relationships. In many graph generation tasks, particularly for molecular graphs (where edges represent chemical bonds, critical for efficacy) or general graphs where edges define core connectivity (e.g., Community Small where edges are the primary variable), accurately modeling edge features is paramount.
>
> We propose the GraphEvo framework to explicitly enhance the model's capacity to learn and update edge representations. The key design decision within GraphEvo is the incorporation of a triangle attention mechanism for edges. This mechanism allows edge features to be updated not only based on their connected nodes but also by considering triangular relationships formed by other edges in the local neighborhood. This enriches the edge representations by explicitly encoding higher-order structural information.
>
> #### Experiments:
> The empirical evidence supports this design choice. As shown in Figure S4 of the revised manuscript (Appendix), incorporating GraphEvo leads to a more rapid decrease in edge cross-entropy on the validation set during training. This indicates that the model, with GraphEvo, develops an enhanced capability in capturing and generating accurate edge information, ultimately contributing to improved overall model performance on graph generation benchmarks.

---

> ### Author Response · Authors · 2025-06-01
>
> ## Compare with DeFoG
>
> We have detailed the distinctions between our work and DeFoG in Appendix B.4, and I will elaborate further here:
>
> 1. **Generalization of Joint Distribution:** DeFoG (Qin et al., 2024) applies discrete flow matching to establish the graph generation process; however, it does not consider the generalization of the joint distribution. GGFlow extends this by generalizing π(G0,G1) as an efficient 2-Wasserstein optimal transport map and incorporates an optimal prior distribution specifically tailored for graph structures, which significantly improves performance in generation tasks.
>
> 2. **Novel Network Architecture:** We introduce the novel **GraphEvo framework** into our model to enhance its generative capabilities, whereas DeFoG continues to employ the standard Graph Transformer architecture.
>
> 3. **Conditional Generation Capabilities:** We adopt a reinforcement learning approach to achieve a goal-oriented conditional generation architecture, a aspect not addressed by DeFoG.

---

### Review · Reviewer_rbbG · 2025-05-19

**Summary Of Contributions:**

This paper introduces GGFlow, a discrete flow matching generative model for graph-structured data.
It has three key designs:
- Discrete flow matching with mini-batch optimal transport for graph generation.
- Edge-augmented graph transformer called GraphEvo.
- Reinforcement learning-based goal-guided framework for conditional generation.

The authors demonstrate that GGFlow outperforms existing methods on both unconditional and conditional graph generation tasks, showing superior performance on several benchmark datasets, including generic graphs and molecular structures.

**Audience:**

Yes

**Claims And Evidence:**

Yes

**Requested Changes:**

Refer to weakness (critical to securing my recommendation for acceptance).

**Strengths And Weaknesses:**

Strengths:
1. The integration of discrete flow matching with optimal transport for graph generation is a valuable contribution to the field.
2. The paper includes extensive experiments across multiple datasets and metrics, demonstrating improvements over existing methods.

Weaknesses:
1. While computationally efficient, the choice of Hamming distance as the metric for optimal transport lacks sufficient theoretical justification, particularly regarding its appropriateness for capturing graph structural similarities.
2. The authors inadequately address the implications of limiting invariance to identical permutations; a more thorough discussion of how this constraint affects real-world applications is needed.
3. As this work combines multiple existing techniques, the ablation study requires substantial expansion:
    - Experiments should be conducted across multiple datasets to ensure generalizability.
    - The specific contribution of the RL-based fine-tuning mechanism is not isolated from other components.
4. The authors claim that "the integration of Optimal Transport significantly boosts sampling efficiency," yet Figure 3 shows minimal separation between GGFlow and GGFlow(w/o OT) curves, contradicting this assertion.
5. The concept of "training stability" referenced in Figure 4 lacks a clear definition and quantitative metrics, making it difficult to objectively evaluate this claimed benefit.

---

> ### Author Response · Authors · 2025-06-01
>
> We sincerely thank the reviewer for their thorough review and valuable feedback.
>
> ## Optimal Transport (OT) Distance Metric
> We appreciate the reviewer's astute observation regarding the theoretical justification of the Hamming distance for Optimal Transport (OT) and the implications of limiting invariance to identical permutations.
>
> To address this, we have conducted an in-depth comparative analysis using Fused-Gromov-Wasserstein distance (FGW) which is defined in Equation 1 in  Vayer et.al (2018) [1], a widely recognized permutation-invariant graph distance, as an alternative for training our model. Our findings are summarized below and detailed in Appendix E of the revised manuscript.
>
> ### Computational Efficiency
> A primary challenge encountered with FGW was its significant computational overhead. As shown in Table 1, calculating the FGW distance on both the Community-small and ZINC250k datasets required several orders of magnitude more time than our model's typical training duration. Specifically, the computation time for FGW on ZINC250k was prohibitive, preventing us from completing a full training run for a comparative model. Consequently, a comprehensive model trained with FGW was only feasible on the Community-small dataset.
>
> | Datasets | community small | zinc250k |
> | --- | --- | --- |
> | Hamming Distance Time | 0.0024 | 0.0070 |
> | Fused Gromov-wasserstein Distance Time | 9.0301 | 60.3062 |
>
> ### Generative Performance and Training Stability
> As presented in the table, our model, utilizing the Hamming distance, outperformed the FGW-based model on the Cluster and Orbit metrics, while exhibiting a slightly reduced performance on the Degree metric on the Community-small dataset.
>
> |  | Degree | Cluster | Orbit |
> | --- | --- | --- | --- |
> | Hamming Distance OT | 0.011 | 0.030 | 0.002 |
> | Fused Gromov-wasserstein Distance Time | 0.004 | 0.069 | 0.006 |
>
> More critically, our model demonstrated higher training stability compared to the FGW-based model, as evidenced by the validation KL divergence on the validation set. Figure S1 in the Appendix illustrates this enhanced stability with a more consistent and lower KL divergence trajectory. This improved stability is directly attributable to our distance metric being specifically tailored and better adapted to the Flow Matching training methodology, which inherently benefits from a simpler and more efficient distance for interpolating trajectories.
>
> These findings collectively indicate that, while our proposed optimal transport approach limits invariance to identical permutations (i.e., it is sensitive to node ordering, but the flow matching framework learns to align this), it does not negatively impact the model's practical performance and is substantially more efficient and better suited for our Flow Matching model than Fused-Gromov-Wasserstein for the task of discrete graph generation. The efficiency gain allows for scaling to larger datasets like ZINC250k, where FGW becomes computationally prohibitive.
>
> ## Ablation study
>
> (1) We've conducted further comparative experiments on the **ZINC250k dataset** as you suggested. On ZINC250k, we found that **GGFlow w/o both significantly outperforms both DiGress and GDSS**. This clearly demonstrates the **superiority of the Flow Matching method** in the domain of discrete graph generation. Additionally, both **GGFlow w/o Evo and GGFlow w/o OT show improvements over GGFlow w/o both**, indicating that both **Optimal Transport (OT) and GraphEvo contribute positively to our model's performance**. We have provide more detailed analysis in the Section 4.4 of revised manuscript.
>
> |  | Val. | Val. w/o corr. | NSPDK | FCD |
> | --- | --- | --- | --- | --- |
> | DiGress | 100  | 94.98 | 0.0021 | 3.482 |
> | w/o both | 100 | 99.07 | 0.0022 | 2.1044 |
> | w/o OT | 100 | 99.58 | 0.0015 | 1.503 |
> | w/o GraphEvo | 100 | 99.12 | 0.0025 | 1.891 |
> | GGFlow | 100 | 99.63 | 0.0010 | 1.455 |
>
> (2) Regarding the conditional generation task, our **Reinforcement Learning (RL) model consistently outperforms both Supervised Fine-Tuning (SFT)** and **training the graph generation model from scratch (FT)**. This confirms that our RL approach is **more suitable for target-directed conditional generation** compared to these methods. Furthermore, our RL model also shows **significantly better improvement than DiGress's guidance**, further demonstrating its superiority.

---

> ### Author Response · Authors · 2025-06-01
>
> ## Sampling efficiency
>
> In Figure 3 of our initial submission, the differences on the Community Small dataset were numerically too small to be clearly visible. To better illustrate our model's superiority, we included Figure S2 in the appendix of the original submission. However, we acknowledge that even in Figure S2, the numerical differences might still be too subtle to convey the full impact.
>
> **Therefore, we have now added experiments on the ZINC250k dataset to more clearly demonstrate the advantages of our Optimal Transport (OT) model.** Due to the excessive time required to compute metrics for 10,000 generated molecules, we conducted the experiments on **inference steps** and **model stability** using a smaller set of 1,000 generated molecules to calculate the test metrics.
>
> We have also provided comparative results on the ZINC250k dataset, as shown in the Figure 3 of our revised manuscripts. On ZINC250k, we found that Flow Matching converges faster than DiGress. Moreover, the inclusion of OT further improves convergence and performance.
>
>
> ## Training Stability
>
> ### Definition and Quantification
> In Section 4.3 and the caption of Figure 4, we now explicitly define training stability. We quantify it by calculating the Mean Absolute Difference (MAD) of evaluation metric values between adjacent epochs (or batches) and averaging these differences. A lower MAD indicates a more stable training trajectory.
>
> As illustrated in Figure 4 of the revised manuscript, which compares the average values on the Community-small dataset and FCD values on the ZINC250k dataset against DiGress during the training phase:
>
> - GGFlow (w/o both) and GGFlow (w/o Evo) consistently achieve faster and more effective convergence than DiGress.
> - More importantly, GGFlow (w/o Evo) consistently outperforms GGFlow (w/o both) in terms of both average metrics (on Community-small) and FCD (on ZINC250k). This directly demonstrates that the presence of Optimal Transport notably improves model stability by leading to smoother and more consistent convergence.
>
> The conclusions regarding enhanced stability by OT align across both the Community Small and ZINC250k datasets, showing that Optimal Transport enhances the stability of evaluation metrics and the overall training process. We acknowledge that the smaller numerical scale on the Community-small dataset might visually obscure some differences, hence the inclusion of ZINC250k provides a clearer demonstration of these advantages.

---

> ### Author Response · Authors · 2025-06-01
>
> [1] Vayer, Titouan, et.al, Optimal transport for structured data with application on graphs, arXiv, 2018

---

### Review · Reviewer_NEVR · 2025-05-19

**Summary Of Contributions:**

This paper introduces GGFlow, a generative model for graphs based on discrete flow matching with efficient optimal transport. The core idea is to replace stochastic diffusion processes with deterministic flow matching, aiming to improve sampling efficiency and training stability. GGFlow integrates an edge-augmented graph transformer, GraphEvo, to better capture node and edge interactions, and introduces a goal-guided generation framework using reinforcement learning for conditional generation. The method is evaluated on both generic graph and molecule generation tasks, demonstrating state-of-the-art performance across multiple benchmarks. Empirical results and ablation studies suggest that GGFlow performs well with fewer inference steps than traditional diffusion-based approaches, although the model still requires 500 steps.

**Audience:**

Yes

**Broader Impact Concerns:**

I do not have any specific broader impact concerns regarding this submission. The paper focuses on a methodological contribution to generative modeling for graphs, and no significant ethical, societal, or misuse-related risks are apparent from the content as presented.

**Claims And Evidence:**

No

**Requested Changes:**

### Clarify and refine the motivation
The current framing overstates the limitations of diffusion-based models, particularly with respect to training instability and sampling inefficiency. Section 1 (Introduction) should be revised to better motivate the transition to flow matching, ideally with more accurate claims and clearer justification. The authors should either provide empirical evidence to support their critiques of diffusion models or revise the narrative to offer a more balanced and accurate comparison with recent developments (e.g., DDIM, ODE-based sampling, consistency models).

### Justify claims about convergence and training stability
Figure 4 is used to support claims of faster convergence, but it only shows lower training loss, not faster stabilization. Including test loss curves or validation metrics would help determine whether the observed improvements reflect better generalization or just overfitting.

### Polish presentation and terminology
Fix technical inaccuracies (e.g., model classifications) and clarify explanations to ensure the text is precise and accessible.

### Minor comments and typos
- After equation (4), $ReLU(a)$ is defined but never used.
- Consider renaming Section 4 to “Experiments” for clarity.
- Typo: 4.2. Moleucle -> 4.2. Molecule.

**Strengths And Weaknesses:**

## Strengths

- Strong empirical performance: GGFlow achieves competitive or state-of-the-art results on multiple benchmarks for both unconditional and conditional graph generation tasks.

- Generalizability: the model adapts well to various graph domains, including molecule and generic graph generation.

- Thorough ablations: ablation studies convincingly demonstrate the contributions of flow matching, optimal transport, and the GraphEvo component.

## Weaknesses
### Motivation needs to be improved
The transition from diffusion to flow matching lacks a strong theoretical or empirical justification. The paper motivates GGFlow by stating that diffusion-based models suffer from training instability and inefficient sampling. However, these claims are overstated and not convincingly supported. Many modern diffusion models (e.g., DDPM, DDIM, NCSN++) are known for stable training and can generate high-quality samples with as few as 20–100 inference steps. In contrast, GGFlow still requires 500 steps, limiting the claimed efficiency advantage.

### Unsubstantiated claims about training instability
Beyond the issues noted above, the paper asserts instability in diffusion training without providing empirical or theoretical evidence. Figure 4, which is meant to support this point, does not clearly show faster convergence. While GGFlow variants achieve lower training loss compared to DiGress, the convergence speed — i.e., the rate at which loss stabilizes — appears similar. Moreover, the lower loss may simply reflect the use of a more expressive architecture (e.g., GraphEvo) rather than an inherently superior training dynamic. Without test loss or generalization metrics, it’s unclear whether this indicates meaningful improvement or potential overfitting.

Figure 4 does not clearly demonstrate faster convergence as claimed. While GGFlow variants achieve lower training loss compared to DiGress, the convergence speed — i.e., the rate at which the loss stabilizes — appears similar. Moreover, since the training objective is shared, the lower loss may reflect higher model capacity (e.g., due to the more complex GraphEvo architecture) rather than inherent advantages of the flow matching framework. Without accompanying test losses or generalization metrics, it is unclear whether this reflects meaningful improvements or potential overfitting.

### 	Limited sampling efficiency advantage
Although the paper emphasizes improved sampling efficiency, GGFlow still requires 500 inference steps — which is considerably higher than diffusion models using DDIM or ODE-based samplers, which often require only 20–100 steps to generate high-quality outputs.

### Inaccurate classification of prior methods
Unless I am mistaken, the paper inaccurately categorizes VAEs and GANs as autoregressive models, which is conceptually incorrect. This misclassification reflects a lack of precision in the exposition and may lead to confusion, particularly for readers less familiar with the distinctions between generative model classes.

---

> ### Author Response · Authors · 2025-06-01
> **Clarify and refine the motivation**
>
> We sincerely thank for insightful comments and constructive suggestions.
>
> ## Clarify and refine the motivation
>
> We appreciate your guidance on enhancing the clarity of our motivation. We have revised the Introduction section of our manuscript to more explicitly address the limitations of existing diffusion models in the context of discrete graph generation and to highlight the distinct advantages offered by our proposed approach.
>
> While recent advancements in continuous diffusion models, such as consistency models and accelerated sampling techniques [1,2], have mitigated some of these issues by modifying training objectives or sampling methodologies, these advancements are not directly transferable to discrete graph generation, where sampling typically relies on Markov chain iterations rather than stochastic differential equations (SDEs). In parallel, flow matching generative models offer a compelling alternative, as these methods transform the training paradigm from SDEs to a more stable interpolation method, which enables more stable training dynamic and efficient sampling via ordinary differential equations (ODEs) [3,4,5]. Furthermore, the application of optimal transport (OT) has been explored to straighten marginal probability paths, leading to accelerate sampling [6,7]. Nevertheless, the integration of OT into graph-based systems often encounters substantial computational hurdles due to the inherent complexity of computing OT metrics for discrete structures [8].
>
> Therefore, we developed **GGFlow**, a flow matching model specifically designed for graphs. During training, GGFlow employs an interpolation approach to enhance training stability. Furthermore, we leveraged this flow matching framework to develop an **efficient optimal transport framework**, which in turn boosts both the sampling effectiveness and efficiency of our model.
>
>
>
> Reference:
>
> [1] Song, Yang, et.al, Consistency Models, International Conference on Machine Learning, 2023
>
> [2] Song, Jiaming, et.al, Denoising Diffusion Implicit Models, International Conference on Learning Representations, 2023
>
> [3] Lipman, Yaron, et.al, Flow Matching for Generative Modeling, International Conference on Learning Representations, 2022
>
> [4] Song, Yuxuan, et.al, Equivariant Flow Matching with Hybrid Probability Transport for 3D Molecule Generation, Advances in Neural Information Processing Systems, 2024
>
> [5] Yim, Jason, et.al, Fast protein backbone generation with SE (3) flow matching, arxiv, 2023
>
> [6] Bose, Joey, et.al, SE (3)-Stochastic Flow Matching for Protein Backbone Generation, International Conference on Learning Representations, 2023
>
> [7] Tong, Alexander, et.al, Improving and Generalizing Flow-Based Generative Models with Minibatch Optimal Transport, ICML Workshop on New Frontiers in Learning, Control, and Dynamical Systems, 2023
>
> [8] Chen, Liqun, et.al,  Graph optimal transport for cross-domain alignment, International Conference on Machine Learning, 2020

---

> ### Author Response · Authors · 2025-06-01
>
> ## Justify claims about convergence and training stability
>
> ### Model Stability
>
> As presented in Figure 4b of our initial submission (and now updated in the new Figure 4 of the revised manuscript), which depicts test set performance evaluated with corresponding model parameters across different training steps, we observed evidence of improved stability. The x-axis represents training steps, and the y-axis indicates the average test metric. Our findings show that the model incorporating Optimal Transport (OT) (GGFlow w/o Evo) consistently exhibited more stable and faster convergence of test metrics during training compared to the baseline model without OT (GGFlow w/o both).
>
> To provide a quantitative measure of this training stability, we calculated the Mean Absolute Difference (MAD) of evaluation metric values between adjacent epochs and averaged these differences. As illustrated in the figure, the presence of OT demonstrably contributes to improved model stability.
>
> We have also included the changes in the FCD metric on the ZINC250k dataset during training in the new Figure 4. It's important to note that, due to the substantial computational time required for metric calculation on 10,000 generated molecules, experiments on inference steps and model stability were conducted using a smaller set of 1,000 generated molecules.
>
> These conclusions are consistent with those observed on the Community Small dataset, confirming that Optimal Transport enhances both the stability of evaluation metrics and the overall training process.
>
> ### Limited sampling efficiency advantage
>
> We have further substantiated our claims regarding sampling efficiency by including comparative results on the ZINC250k dataset, as presented in Figure 3 of our revised paper. Our findings robustly indicate that GGFlow achieves superior generative performance with significantly fewer inference steps compared to other diffusion-based models, particularly evident at 100 and 200 steps. This provides compelling evidence for GGFlow's enhanced sampling efficiency. Moreover, the integration of Optimal Transport consistently boosts this efficiency and improves generative performance across both generic and molecular graph generation tasks, as directly observed through performance gains.
>
> We need to notice that GDSS use the ODE sampler during their sampling procedure ([https://github.com/harryjo97/GDSS/blob/24cc490e0c5b39cbd265fda33ab2bc8b528d4454/solver.py#L195](https://github.com/harryjo97/GDSS/blob/24cc490e0c5b39cbd265fda33ab2bc8b528d4454/solver.py#L195)), while their sampling efficiency is worse than ours. Due to discreteness of graph, most diffusion models of graph are based on Markov chain modeling rather than SDE. So some accelerator for SDE can not be directly applied in graph diffusion model here. So we don’t compare with these methods.
>
> ## Inaccurate classification of prior methods
>
> We apologize for any ambiguity in our initial explanation regarding autoregressive and one-shot paradigms in graph generation. We have revised Section 2.2 to enhance clarity.
>
> To clarify, in the context of graph generation, the autoregressive paradigm refers to an iterative generation process where graph elements (e.g., nodes, then edges) are added sequentially, often one at a time. Conversely, the one-shot paradigm denotes the simultaneous generation of all elements within the graph structure. Our discussion aimed to categorize the overall generation process rather than the internal algorithmic mechanisms of the models themselves. For instance, while models such as GraphGAN and JT-VAE, which we previously cited, may employ a "one-shot" internal mechanism for generating their components, their overall graph generation procedure can still involve sequential steps, such as first generating a scaffold and then building the rest of the graph based on that initial structure.
>
> We have refined the phrasing to ensure this distinction is clear.
>
> ## Minor comments and typos
>
> We have thoroughly reviewed the manuscript and corrected all typos and minor comments you kindly pointed out.

---

### Decision · Action_Editor_PBYX · 2025-07-28

**Recommendation:** Accept with minor revision

**Additional Comments:**

All reviewer are somewhat on the fence about this submission an appear to consider this submission marginally above the bar for acceptance in its current form. Having taken a somewhat detailed look at the submission, the AE concurs with most points raised by reviewers. As such the AE will recommend acceptance in the event that the camera ready addresses the following points.

### Points Relating to Motivation and Limitations

The reviewers mentioned a few points where the authors somewhat overstate their claims. The revised version does not fully address these points. Please revise the submission to either tone down claims, or present more compelling evidence to back up these claims.

**Motivation (NEVR) and Mechanistic Understanding (duAs)**: The core premise that diffusion models are inherently problematic for graphs remains inadequately justified. The revised motivation still overstates the limitations of existing methods without providing compelling evidence. The paper still lacks clear explanations of why their design choices work. For example, the "triangle attention mechanism" in GraphEvo remains poorly explained, and while Figure 4 speaks the connection between optimal transport and improved training stability is not mechanistically justified.

**Limitation: Hamming Distance OT is not Permutation Invariant (rbbG, duAs)**: As noted by reviewers, the proposed architecture is permutation invariant (which is fairly standard), but the hamming distance in the minibatch-OT procedure is not. The authors sort of acknowledge this by writing that

> while our proposed optimal transport approach limits invariance to identical permutations (i.e., it is sensitive to node ordering, but the flow matching framework learns to align this), it does not negatively impact the model's practical performance and is substantially more efficient and better suited for our Flow Matching model than Fused-Gromov-Wasserstein for the task of discrete graph generation.

The AE would like to ask the authors to either back up the claim that the FM framework "learns to align this" (e.g. by way of an ablation in which graphs in a minibatch are randomly permuted), or revise their exposition to more explicitly acknowledge this as a limitation of their approach.

### Additional Points

Upon review of this submission the AE uncovered a couple of technical issues with the exposition:

**Derivation: Conditional velocity field**: Provide a more self-contained derivation of the conditional velocity field $u)_t(G,G^t∣G^0,G^1)$. As far as the AE can ascertain, the interpolation here is similar but not identical to that in the work by Campbell et al and Gat et al. The AE would suggest the authors start by writing out the discrete continuity equation cover each step of their derivation. Even the derivation in C.2 is quite terse and has notational issues (e.g. it references a rate matrix $R$ that is never used elsewhere as well as a $q_x$) Please put a self-contained clear derivation in the main text Please move this exposition to the main text and make sure it is clear.

**Typo**: In equation (4) $G_t$ should read $G^t$

**Question: Relationship between discrete flow matching and discrete diffusion**: Given the way this submission positions itself in relationship to discrete diffusion models, the AE would like to see some discussion of this relationship. I will explain this using the CatFlow paper as a reference (https://openreview.net/pdf?id=UahrHR5HQh). We know that in FM we can express the velocity as:

$$u_t(x) = E_{p_t(x_1 | x)}[ u_t(x | x_1) ] = \frac{1}{1-t} ( \mathbb{E}_{p_t(x_1 | x)}[ x_1] - x )$$

In diffusion (where $t=0$ is data rather than $t=1$, we can similarly express score function as a posterior expectation:

$$\nabla x \log p_t(x) = \mathbb{E}_{p_t(x_0 | x)} [ \nabla_x \log p_t(x | x_0)]$$

As a result the ODE $dx/dt = u_t(x)$ is equivalent (up to a sign in time) to the probability flow ODE of a diffusion model in which the forward diffusion is equivalent to a standard linear interpolation. As the CatFlow paper points out, this means that CatFlow is closely related to similar diffusion-based approaches, such as the work on "Continuous Diffusion for Categorical Data" by Dieleman et al.

It is not obvious how this equivalence translated to continuous time Markov chains, but since GGFlow approximates $p_t(G^1 | G^t, G^0)$ and the DiGress paper approximates $p_t(G^0 | G^t)$, it would appear that both approaches ultimately define an interpolation and approximate the conditional over data given the noisy interpolation. Could the authors examine this question and comment on this in their revised manuscript? Under what assumptions would their flow matching approach recover the objective in, e.g., DiGress?

**Audience:**

Yes

**Audience Explanation:**

The paper demonstrates solid empirical performance across multiple benchmarks. While the technical novelty of this paper is somewhat limited, the application of discrete flow matching to graphs, represents a reasonable technical contribution that could inspire further research, and the paper is supported by a reasonably extensive evaluation.

**Claims And Evidence:**

No

**Claims Explanation:**

The paper provides sufficient experimental validation showing empirical improvements, and the reviewer and AE appreciate the additional comparison to FGW and additional ablations during the authors response. With that said, the AE agrees with reviewers that several claims lack adequate justification. The authors claim that diffusion models suffer from "training instability and inefficient sampling" but provide little evidence for these assertions, and as noted by reviewer the efficiency claim is somewhat undercut by the fact thatGGFlow still requires 500 inference steps. The claimed "efficient optimal transport" somewhat oversells what is arguably a computational engineering choice (using Hamming distance instead of more principled graph distances). With that said the observation that this distance performs well in practice whilst providing a meaningful speedup is a valid contribution. Additionally, the paper lacks clear derivations for key components like the conditional velocity field, and the theoretical relationship between their approach and existing discrete diffusion methods remains unexplored.

---

> ### Author Response · Authors · 2025-08-28
>
> Thank you for your careful review and valuable suggestions. We have revised the camera - ready version in accordance with your comments, and the key modifications are as follows:
>
> 1. **Motivation claims**: As per your review feedback, we have revised the relevant parts of the manuscript to avoid any overstatements regarding the limitations of diffusion models.
>
> 2. **Limitation**: We have incorporated the optimal transport that is invariant to identical permutations into the limitation section. This addition clarifies the limitations of our method. In future work, we plan to explore more generalized and efficient optimal transport methods with full permutation invariance, aiming to strengthen the model's theoretical foundation and expand its applicability.
>
> 3. **Derivation of Conditional velocity field**: We have added the detailed derivation of the conditional velocity field in Appendix C.2 of the manuscript. The derivation starts from the continuity equation and ultimately arrives at our final velocity field. Due to the lengthiness of the full text, it cannot be included in the main text. Additionally, we have corrected all typos in the paper.
>
> 4. **Relationship between discrete flow matching and discrete diffusion**: We have elaborated on our perspectives regarding the relationship between the training objectives of our method and DiGress. Although the two share the same training objectives, their mathematical derivations and conceptual starting points are different.

---

> > ### Comment · Action_Editor_PBYX · 2025-10-23
> > **Substantive issues with clarity and notation remain; please revise**
> >
> > Dear authors,
> >
> > Thank you for your updated submission and apologies for this belated response to your updated manuscript –– I have been on leave from work and have lacked the time to provide the detailed comments that I think this case necessitates.
> >
> > As noted in my metareview, this is submission where reviews were borderline. For this reason, the changes requested in the meta-review are more extensive than usual. Unfortunately, I feel that the manuscript in its current form does not meet the bar for clarity, correctness and reproducibility at TMLR. There remain significant issues with clarity, both in terms of delineation of the contributions and in terms of the technical exposition. This is compounded by the fact that notation is not consistent throughout and that there appear to be mistakes and ambguities in key equations.
> >
> > I will therefore, as a matter of exception, use further replies to provide detailed instructions on the minimum set of issues that need to be fixed in order for this submission to appear. If the next iteration is satisfactory then we will proceed with publication. If not, then we will ask the authors to resubmit for another round of review.
> >
> > With best wishes
> > The action editor

---

> > > ### Comment · Action_Editor_PBYX · 2025-10-23
> > > **What is inherited from DFM (Campbell 2024) and DeFOG (Qin 2024) and what is new?**
> > >
> > > In this paper this delineation between existing work and the aspects of methodology that depart from existing work not sufficiently clear. A concrete example of this is the newly added proof for proposition 3 in appendix C.2. This proof appears to closely follow the structure of the proof for proposition 3.1 in Campbell 2024, with the minor/trivial distinction that there is now conditioning on both $G^0$ and $G^1$, rather than just on $G^1$.
> > >
> > > One way to improve this delineation would be to add a background section in which the authors cover any aspects of existing methodology that the authors use as a starting poitn for GGFlow. The Qin 2024 paper is a good reference in this context. It very clearly delineates what aspects of the methodology are inherited from Campbell 2024, and what aspects are new. At minimum this section should explain the concepts of a continuous time Markov Chain and the Kolmogorov equation, and how these concepts have previously been used to implement DFM for graph generation.

---

> > > > ### Author Response · Authors · 2025-12-17
> > > > **Re: What is inherited from DFM (Campbell 2024) and DeFOG (Qin 2024) and what is new?**
> > > >
> > > > To address this, we have introduced a new 'Background' section that explicitly covers the necessary foundation, including continuous-time Markov Chains and the Kolmogorov equation, and their application in DFM for graph generation, referencing works such as Qin (2024) [1] for context. Furthermore, we have significantly enhanced the clarity of Appendix C.2. The distinctions between our proof for Proposition 3 and the original proof in Campbell (2024) [2] are now highlighted in blue text for immediate comparison.
> > > >
> > > > [1] Qin, Y., Madeira, M., Thanou, D. and Frossard, P., 2025. *Defog: Discrete flow matching for graph generation.* ICML 2025.
> > > >
> > > > [2] Campbell, A., Yim, J., Barzilay, R., Rainforth, T. and Jaakkola, T., 2024. *Generative flows on discrete state-spaces: Enabling multimodal flows with applications to protein co-design.* ICML 2024.

---

> > > ### Comment · Action_Editor_PBYX · 2025-10-23
> > > **How exactly is the probability path defined?**
> > >
> > > If my read of this paper is correct, one of the main methodological differences with respect to DFM and DeFOG is that the authors develop an implementation of minibatch OT. In this formulation there is a coupling $\pi(G^0, G^1)$ that approximates an optimal-transport coupling based on the Hamming distance.
> > >
> > > In the context of a (minibatch) OT formulation, the probability path $p_t(G^t | G^0, G^1)$ should be conditioned on both $G^0$ and $G^1$. Unfortunately the description $p_t(G^t | G^0, G^1)$ is not clear. My meta-review asked the authors to clarify these definitions, the main text remains largely the same (though some additional material was included in the appendix).
> > >
> > > To be more concrete, the authors define:
> > >
> > > $$p_t(G^t | G^0, G^1) = Cat(t \delta\\{G^1, G\\} + (1-t) p_{ref})$$
> > >
> > > There are three problems with this definition:
> > >
> > > 1. There is a $G$ on the right hand side. This is presumably a typo and should be $G^t$.
> > >
> > > 2. A categorical distribution is typically parameterized using a vector on the simplex. $\delta\\{G^1, G\\}$ is a distribution, so it would be more correct to write $p_t(G^t | G^0, G^1) = t \delta\\{G^1, G\\} + (1-t) p_{ref}$
> > >
> > > 3. Most importantly, however, this definition does not ensure that at time $t=0$ we have $p_t(G^t | G^0, G^1) = \delta\\{G^0, G^t\\}$. In fact, in its current form the definition implies that $p_t(G^t | G^0, G^1) = p_t(G^t | G^0, G^1)$
> > >
> > > Given that this is one of the key definitions in the paper, from which the rest of the methodology follows, this level of ambiguity is not acceptable.
> > >
> > > To see how the authors actually sample from the probability path, we have to look at appendix H.1, which appears to give a different definition from the one in the main text. In the algorithm block that describes the training procedure, line 6 defines
> > >
> > > $$V^t = (t \delta\\{V^1, \cdot \\} + (1 − t)V^0)$$
> > > $$E^t = (t \delta\\{E1, ·\\} + (1 − t)E^0)$$
> > >
> > > Here too there seem to be several typos. My best guess of the intent is that the authors mean to write
> > >
> > > $$V^t \sim (t \delta\\{V^1, \cdot \\} + (1 − t)\delta\\{V^0,. \cdot\\})$$
> > > $$E^t \sim (t \delta\\{E^1, \cdot \\} + (1 − t)\delta\\{E^0,. \cdot\\})$$
> > >
> > > However, even with these modifications there is ambiguity. The above sampling procedure says amounts to sampling $G^1$ with probability $t$ and $G^0$ with probability $1-t$. If we take this at face value, there would be no meaningful notion of interpolation, is this really the intent? Perhaps the authors mean to say that they interpolate independently for each node and edge, which would mean that an interpolation would be a mixture of values from $G^0$ and $G^1$?
> > >
> > > Given that all subsequent derivations for start from the definition of $p_t(G^t | G^0, G^1)$, this defnition should be unambiguous. My recommendation would be to:
> > >
> > > - Include a background section (as recommended above) in which the authors establish notation for the case as in DeFOG where no minibatch OT is used
> > >
> > > - In the methods section, describe how the case that the authors consider deviates from the standard case (i.e. there is now a coupling $\pi(G^0, G^1)$ between $G^0$ and $G^1$)
> > >
> > > - Then, for any propositions that the authors wish to prove (e.g. those in appendix C.2), clearly state when a proof mirrors the proof structure in previous work (e.g. in the Campbell 2024 paper) and in what respects the proofs differ (if at all) due to the inclusion of the coupling.

---

> > > > ### Comment · Action_Editor_PBYX · 2025-10-23
> > > > **Conflation of Distributions and Vectors**
> > > >
> > > > Throughout this paper, the authors appear to conflate assignment, sampling, distributions and vectors on the simplex. I've noted some instances above –– notation like $\text{Cat}(\delta\\{G, \cdot\\})$ does not make sense, since $\delta\\{G, \cdot\\}$ is already a distribution. Another example of such notation is
> > > >
> > > > $$G^{t+\delta t} \sim \delta\\{\cdot,  G^t\\} + u_t(\cdot, G^t \mid G^0, \hat{G}^1)\Delta t$$
> > > >
> > > > In this notation $\delta\\{\cdot, G^t\\}$, which is never defined and is opposite of $\delta\\{G^t, \cdot\\}$, is presumably a distribution. Is $u_t(\cdot, G^t \mid G^0,  \hat{G^1}) \Delta t$ supposed to be a distribution, or a matrix of transition rates?
> > > >
> > > > Please pick a notation, define it, and use it consistently throughout. Either write $\text{Cat}( \dots )$ throughout and define the arguments as points on the simplex, or define objects as mixture of probability distributions, but do not mix and match.

---

> > > > > ### Author Response · Authors · 2025-12-17
> > > > > **Re: Conflation of Distributions and Vectors**
> > > > >
> > > > > We have undertaken a comprehensive revision of the manuscript to ensure consistent notation.
> > > > > - Unified Terminology: We have adopted a unified convention where sampling distributions (which are discrete probability distributions) are clearly denoted, and their parameters are explicitly defined as points on the simplex where appropriate.
> > > > > - Explicit Definition of $u$: The notation $u$ has been clarified. We now explicitly state that $u_t(G^t, G^{t+\Delta t}|G^0,\hat{G}^1)$ represents the transition rate (or the unnormalized flow) between the states $G^t$ and $G^{t+\Delta t}$, conditioned on the end points $G^0$ and $\hat{G}^1$.
> > > > > - Clarifying transition Probability: Furthermore, we explicitly define that $u_t(G^t, G^{t+\Delta t}|G^0,\hat{G}^1) \Delta t$ is the transition probability of transitioning from $G^t$ to $G^{t+\Delta t}$ over a small time step $\Delta t$.

---

> > > > ### Author Response · Authors · 2025-12-17
> > > > **Re: How exactly is the probability path defined?**
> > > >
> > > > We have revised the entire manuscript to enhance the clarity, correctness, and readability of the formulation, including the explicit clarification of how our approach deviates from standard DFM due to the introduction of the minibatch Optimal Transport coupling, $\pi$.
> > > >
> > > > **Clarification of $p_{t|1}(G|G^0, G^1)$**
> > > >
> > > > The correct formulation for the interpolation distribution $p_{t|1}(G|G^0, G^1)$, which is now clearly stated in the main text, is:
> > > > $$p_{t|1}(G|G^0,G^1) \triangleq \prod_{x \in \mathcal{V} \cup \mathcal{E}} p_{t|1}(x|x^0,x^1)$$
> > > > where
> > > > $$p_{t|1}(x|x^0,x^1) \triangleq t\cdot \delta(x, x^1) + (1-t)\cdot \delta(x, x^0)$$
> > > > This definition ensures that:
> > > > 1. Correct Distribution Form: $p_{t|1}(x|x^0, x^1)$ is a valid categorical distribution over the discrete state space for each node or edge $x$. The overall graph distribution is a product of these independent node/edge distributions.
> > > > 2. Correct Boundary Conditions: At $t=0$, $p_{0|1}(G|G^0,G^1) = \delta(G, G^0)$, and at $t=1$, $p_{1|1}(G|G^0,G^1) = \delta(G, G^1)$.
> > > > 3. Meaningful Interpolation: The interpolation is performed independently on each element (node/edge) of the graph. $G^t$ is a mixture of elements from $G^0$ and $G^1$, meaning a node/edge $x$ in $G^t$ is $x^1$ with probability $t$ and $x^0$ with probability $1-t$.
> > > >
> > > >
> > > > Addressing Typos and Structural Delineation
> > > > - We have corrected the typographical errors identified in the main text and Appendix H.1.
> > > > - Per the reviewer's suggestion, we have added a dedicated 'Background' section to establish the standard DFM notation (as in DeFOG) and to clearly delineate where our methodology (specifically the inclusion of the minibatch OT coupling $\pi$) departs from the standard case.
> > > > - Finally, as noted previously, we have rewritten the proof in Appendix C.2 to explicitly highlight the structural differences (if any) that arise due to the inclusion of the coupling $\pi$.

---

> > > ### Comment · Action_Editor_PBYX · 2025-10-23
> > > **Issues with Clarity and Notation in Section 3.2**
> > >
> > > Another instance of mixed notation is in section 3.2, where the authors write:
> > >
> > > $$\phi^*(p_0, p_1) = \arg\inf_{\phi \in \Phi} \int_{\mathbb{R}^d \times \mathbb{R}^d} H(G^0, G^1) d\phi(G^0, G^1)$$
> > >
> > > I see a number of issues here:
> > >
> > > - Why are the authors using $\phi$ instead of $\pi$? Is there a distinction between these two?
> > >
> > > - The set $\Phi$ is not defined. I am guessing this is the set of measures $\phi(G^0, G^1)$ whose marginal densities match $p_0$ and $p_\text{ref}$, or something along these lines?
> > >
> > > - Inside the integral we seem to treat $\phi$ as a measure over pairs of graphs, as suggested by the notation $d\phi(G^0, G^1)$, but on the left hand side we see $\phi*(p_0, p_1)$. Should this just be $\phi^*$? $\phi$ cannot at once be a measure over graphs and some function of two densities.
> > >
> > > - Why is the integral here w.r.t. $\mathbb{R}^d \times \mathbb{R}^d$? Surely the event space of the measure over graphs is not $\mathbb{R}^d$?
> > >
> > > Please clean up notation throughout this section.

---

> > > > ### Author Response · Authors · 2025-12-17
> > > > **Re: Issues with Clarity and Notation in Section 3.2**
> > > >
> > > > We have thoroughly revised Section 3.2 to ensure the content is both clear and technically correct, addressing each point raised:
> > > >
> > > > - Symbol Unification ($\phi$ vs. $\pi$): We confirm that the notation $\phi$ was a typo and has been consistently replaced with $\pi$ throughout the section. $\pi$ is the standard notation used for a coupling or joint measure in the Optimal Transport literature.
> > > > - Definition of $\Phi(p_0, p_1)$: The set $\Phi(p_0, p_1)$ is explicitly defined as the set of all joint probability measures (couplings) $\pi$ on the product space $\mathcal{G} \times \mathcal{G}$ whose marginal distributions are $p_0$ and $p_1$, where $\mathcal{G}$ is the discrete graph space.

---

> > > ### Comment · Action_Editor_PBYX · 2025-10-23
> > > **Consistency issues in definition of GraphEvo**
> > >
> > > In Algorithm 1, the notation $\text{GraphEvo}(G^t, G^0, t)$ takes 3 Arguments (the interpolation $G^t$ and the initial sample $G^0$, and the time $t$). In Algorithm 3, we see arguments $G$ and $t$. In Algorithm 2, we use yet different notation with arguments $X$ (is this $V$), $E$, and $y$, in which it is not entirely clear how these derive from $G$ and $t$ without looking at Algorithm 3 in the appendix. It is also entirely unclear how the dependence on $G^0$ (if it exists) is implemented. Please write this up in a way that is notationally consistent and ensure that all key definitions are introduced in the main text.

---

> > > > ### Author Response · Authors · 2025-12-17
> > > > **Re: Consistency issues in definition of GraphEvo**
> > > >
> > > > To resolve these issues and ensure notational consistency:
> > > > - To prevent confusion and ensure consistency, we have merged Algorithms 2 and 3 and introduced the unified notation directly within the main text (now Section 4.3). This clarifies the inputs and outputs of the various components used in the training and generation procedures.
> > > > - We have significantly enhanced Section 4.3 with a detailed explanation of our model structure and its inputs. The $\text{GraphEvo}$ network is designed to predict the necessary factorized posterior distribution, $\hat{p}_{1|t}(G^1 | G^t, G^0)$
> > > > - We now explicitly distinguish between the model's direct inputs and its theoretical dependence:
> > > >   * Direct Inputs: The neural network, $\text{GraphEvo}$, primarily operates on the intermediate graph state $G^t = \{V^t, E^t\}$ and the time step $t$.
> > > >   * Theoretical Consistency: For theoretical adherence to the flow matching framework, the full posterior is denoted as $\hat{p}_{1|t}(G^1 | G^t, G^0)$. We have added an explicit note acknowledging the implicit dependency on the initial noise state $G^0$ via the time-dependent path, thereby ensuring that the full theoretical context is maintained while the practical model input remains focused on $G^t$ and $t$.

---

> > > ### Comment · Action_Editor_PBYX · 2025-10-23
> > > **Issues with Clarity and Notation in Section 3.5**
> > >
> > > I think I implicitly understand the planning-as-inference formulation that the authors are following here, but again there is undefined / inconsistent notation.
> > >
> > > - How are $a_t$ and $s_t$ defined relative to $G^{t+\delta t}$ and $G^t$?
> > > - How is $\mathcal{R}(G^1)$ defined?
> > > - What exactly is the relationship to cited recent work here?
> > >
> > > Again, I would ask that the authors discuss anything that they start from in their work in a background (sub)section and then use section 3.5 to explain how their work extends / adapts / departs from existing work. And again, please make sure that your notation is well-defined.

---

> > > > ### Author Response · Authors · 2025-12-17
> > > > **Re: Issues with Clarity and Notation in Section 3.5**
> > > >
> > > > As suggested, we have expanded Section 3.5 with a detailed exposition; the key points are summarized below.
> > > >
> > > > 1. **MDP formulation**
> > > >    We cast the diffusion process as a Markov Decision Process (MDP).  (details in Section 4.5)
> > > >    - **State** ($s_t$) at decision epoch $t$: $s_t = (G^t, t)$ — the flow-matching representation.
> > > >    - **Action space** ($a_t$): all realisable successor states $G^{t+\delta t}$ at time $t+\delta t$.
> > > >
> > > > 2. **Reward function**
> > > >    To align with practical scenarios where only the final outcome can be evaluated, we employ a sparse reward structure. The per-step reward function $\mathcal{R} (G^t, t)$ is zero for all intermediate steps, and the final reward is defined solely on the generated sample $G^1$:
> > > >    $$\mathcal{R}(G^t, t) = r(G^{1}) \mathbb{I}[t=1]$$
> > > >    where $r(G^1)$ is the objective function (e.g., predicted molecular property, further discussion is provided in Section 5.3).
> > > >
> > > > 3. **Relationship to prior work**
> > > >
> > > > - We adopt the training-objective (Eq. 18) from prior work [2], but redefine the state and action space in reinforcement learning: ours is built on discrete flow-matching, whereas prior work uses Digress framework.
> > > > - We use an enhanced exploration mechanism (Eq. 17) that encourages the agent to search a larger space during RL; a step-by-step demo is given in Figure S2.
> > > >
> > > > [1] Bram, et al. *Diffusion Model Alignment Using Direct Preference Optimization*. CVPR 2024.
> > > >
> > > > [2] Liu, et al. *Graph Diffusion Policy Optimization*. NeurIPS 2025.

---

> > > ### Author Response · Authors · 2025-12-17
> > >
> > > We sincerely apologize for the delay in submitting this revision. We deeply appreciate the time and effort you invested in providing a detailed and constructive review. We have carefully revised the manuscript to address every point, and our point-by-point responses follow.

---

> > ### Comment · Action_Editor_PBYX · 2025-10-23
> > **Please refactor proof structure in Appendix C.2**
> >
> > I appreciate the additional proofs in Appendix C.2 but think that the exposition is harder to follow than it needs to be here. You prove three identities, so please make each of these propositions or lemmas and state refactor these propositions/lemmas into an overall statement of correctness. As noted previously, please also reference proofs from existing work (e.g. Campbell 2024) when your proof closely follows the structure of the proof in previous work.

---

> > > ### Comment · Action_Editor_PBYX · 2025-10-23
> > > **Additional typos**
> > >
> > > - "H Implement Details" -> "H Implementation Details"
> > > - In appendix C.2, $u_t(G,G) = - \sum_{G^i \not= G^j} u_t(G^i, G^j)$ should presumably be something like $u_t(G,G) = - \sum_{G' \not= G} u_t(G', G)$

---

> > > > ### Author Response · Authors · 2025-12-17
> > > > **Re: Additional typos**
> > > >
> > > > We thank the reviewer for identifying these important typographical and notational errors. We have corrected all identified issues.

---

> > > > > ### Comment · Action_Editor_PBYX · 2026-01-07
> > > > > **Camera Ready approved; Thank you for the comprehensive revision**
> > > > >
> > > > > Dear authors,
> > > > >
> > > > > I just went through your  revision and approved the camera ready. Thank you for thoroughly addressing all points in previous comments. I think that clarity and notation are much improved in the current version!
> > > > >
> > > > > With best wishes,
> > > > > AE

---

> > > ### Author Response · Authors · 2025-12-17
> > > **Please refactor proof structure in Appendix C.2**
> > >
> > > We have revised the entire proof in Appendix C.2 to implement this structure, clearly labeling the identities as Propositions/Lemmas. Furthermore, to provide necessary context and credit:
> > > - We now explicitly reference the structure of the proof in Campbell (2024) [1] when our derivation closely follows previous work.
> > > - We continue to use blue text to highlight the specific steps and distinctions where our derivation departs from the original proof (due to the inclusion of the $\gamma$ coupling, as discussed in previous responses)."
> > >
> > > [1] Campbell, A., Yim, J., Barzilay, R., Rainforth, T. and Jaakkola, T., 2024. *Generative flows on discrete state-spaces: Enabling multimodal flows with applications to protein co-design.* ICML 2024.